# Relevance-Based Embeddings:
# Lightweight Candidate Retrieval via Heavy-Ranker Calls

**Kirill Shevkunov** [1]  **Andrey Ploskonosov** [1]  **Liudmila Prokhorenkova** [1]

## Abstract

In many machine learning applications, the most relevant items for a query should be efficiently retrieved. The relevance function is usually an expensive similarity model, making the exhaustive search infeasible. A typical solution is to train another model that separately embeds queries and items to a vector space, where similarity is defined via the dot product or cosine similarity. This allows one to search the relevant items through fast approximate nearest neighbor search at the cost of some reduction in quality. To compensate for this reduction, the found items (candidates) are re-ranked by the expensive ranking model. In this paper, we investigate an alternative approach to candidate selection that utilizes the scores of the expensive model to improve the representations of queries and items. The idea is to describe each query (item) by its relevance to a set of support items (queries) and use these new representations to obtain query (item) embeddings. We theoretically prove that such embeddings are powerful enough to approximate any complex similarity model (under mild conditions). We also investigate the choice of support items, which is a crucial ingredient of the proposed approach. The experiments on diverse academic and production datasets illustrate the power of our method.

## 1. Introduction

Finding the most relevant element (item) $i$ to a query $q$ among a large set of candidates $I$ is a key task for a wide range of machine learning problems, for example, information retrieval, recommender systems, question-answering systems, or search engines. In such problems, the final score (relevance) is often predicted by a pairwise function $R : I \times Q \to \mathbb{R}$, where $Q$ is a query space and $R$ approximates some ground truth relevance such as click probability or time spent. Depending on the task, the relevance function $R$ can utilize query attributes (e.g., the text of the query or a set of numerical features describing the user, such as age, time spent on the service, etc.), item attributes, or attributes describing the query-item pair (e.g., statistics based on counts of each query term in the document in information retrieval tasks).

The problem of retrieving the most relevant item for a query $q$ can be written as $\arg\max_{i \in I} R(i, q)$. For practical applications, it is usually required to return not one but $K$ best items (for directly displaying to the user or further re-ranking). Most recommender systems have large item spaces $I$ (millions to hundreds of millions), so exhaustive search is infeasible. This problem is often solved by training an auxiliary model $\tilde{R}$, called a Siamese, two-tower, or dual encoder (DE), in which late binding is used: $\tilde{R}(i, q) = S(F_I(i), F_Q(q))$, where $F_I : I \to \mathbb{R}^d$, $F_Q : Q \to \mathbb{R}^d$, and $S$ is some lightweight similarity measure, usually dot product or cosine similarity.

While a lot of effort has been put into developing dual-encoder models, the cross-encoder (CE) ones are generally more powerful (Wu et al., 2020; Yadav et al., 2022). Moreover, in practice it is typical to also have features that describe a query-item pair: e.g., counts of query terms in the document (information retrieval), information about previous user-item interactions (recommender systems), and so on. Such features cannot be used by dual encoders thus limiting their expressivity.

An alternative approach suggested by Yadav et al. (2022) is to approximate the relevance of a given query to all the items using the relevance of this query to a fixed set of randomly chosen support items. In more detail, the authors apply the matrix factorization to the query-item relevance matrix to represent it as a product of its submatrix containing only a few columns (relevances to support items) and some other, explicitly computable.

Motivated by the above-mentioned work, we propose and analyze the concept of *relevance-based embeddings* (RBE).

---

[1]Yandex. Correspondence to: Kirill Shevkunov <shevkunov@yandex-team.ru>, Liudmila Prokhorenkova <ostroumova-la@yandex-team.ru>.

*Proceedings of the 43rd International Conference on Machine Learning*, Seoul, South Korea. PMLR 306, 2026. Copyright 2026 by the author(s).

The main idea is to describe queries by their relevance to some pre-selected support items and describe items by their relevance to some support queries. Then, such representations can be used in various ways: as in Yadav et al. (2022), they can be multiplied by a certain matrix to obtain relevance approximations, or they can be passed into a neural network to potentially obtain better approximations (trained for a desired loss), or they can be additionally combined with the original node features to get even more powerful embeddings. An important aspect of our approach is how to properly choose support elements. Previous studies (Morozov & Babenko, 2019; Yadav et al., 2022) sampled them uniformly at random, while we show that there is significant room for improvement. We investigate different options: from simple heuristics (e.g., popular or diverse elements) to methods directly optimizing the relevance approximation quality. Surprisingly, even very simple strategies like clustering the elements and choosing the cluster centers as support items already give significant improvements that can be further enhanced by more advanced and theoretically justified strategies.

From a theoretical perspective, we prove that (under mild conditions) relevance-based embeddings are powerful enough to approximate any continuous similarity function. In particular, when the similarity function utilizes pairwise features, dual encoders based on the individual user and item features inevitably lose this information; however, relevance-based representations contain this information and thus can approximate the desired function.

From a practical perspective, our approach uses a significantly smaller number of trainable parameters (compared to the baseline dual encoder candidate selection model) and does not require any feature engineering since the heavy ranker's predictions are used as features. In other words, we propose a method for an effective and efficient candidate selection via any black-box ranker model.

To evaluate the performance of RBE, we conduct experiments on various textual and recommendation datasets. We show that RBE outperforms the methods from Yadav et al. (2022) and Yadav et al. (2024) on publicly available academic datasets. We also demonstrate the advantages of RBE over strong dual encoders in production scenarios by testing RBE as part of two recommendation services.

## 2. Related Work

In this section, we discuss research areas and representative papers related to our study.

The **relevance retrieval problem** is widespread in the development of information retrieval systems (Kowalski, 2007) such as text and image search engines (Huang et al., 2013; Gordo et al., 2016), entertainment recommender

systems (Covington et al., 2016), question answering systems (Karpukhin et al., 2020), e-commerce systems (Yu et al., 2018), and other practical applications.

Usually, such problems are solved by learning **query and item embeddings** into a certain space and then searching for approximate nearest elements in this space, followed by re-ranking via a heavier ranker. Huang et al. (2013); Covington et al. (2016) explicitly use this approach, offering two-tower models (a.k.a. dual encoders). There are also alternatives to dual encoders that use, e.g., BM25 scores applicable to texts (Logeswaran et al., 2019; Zhang & Stratos, 2021) or other approaches (Humeau et al., 2020; Luan et al., 2021). However, there is usually a trade-off between complexity and quality. One can also train the dual encoder by distilling a heavier ranker model (Wu et al., 2020; Hofstätter et al., 2020; Lu et al., 2020; Qu et al., 2021; Liu et al., 2022). Although these works aim at simplifying the learning of light ranking using the heavy one, they differ from our approach since distillation means that there are still two heavy (comparable in orders of magnitude of trainable parameters) models, unlike our lightweight approach. Also, distillation-based approaches do not provide theoretical guarantees similar to those obtained in our work. In particular, if relevance is highly determined by pairwise query-item features, the obtained dual encoder can be weak since it is not able to capture this important signal.

As for the **nearest neighbor search** in a common query-item space, a wide variety of algorithms exist, including locality-sensitive hashing (LSH) (Indyk & Motwani, 1998; Andoni & Indyk, 2008), partition trees (Bentley, 1975; Dasgupta & Freund, 2008; Dasgupta & Sinha, 2013), and similarity graphs (Navarro, 2002). LSH- and tree-based methods provide strong theoretical guarantees but it has been shown that graph-based methods usually perform better (Malkov & Yashunin, 2020; Aumüller et al., 2020), which explains their widespread use in practical applications.

Another research direction is methods that combine nearest neighbor search with heavy-ranker calls (Morozov & Babenko, 2019; Chen et al., 2022; Xu et al., 2025) instead of separately embedding queries and items in a common space where the search for the nearest items can be efficiently performed. Such methods show better quality in comparison with separate embeddings, however, their practical application may be limited since they require a significant change in the structure of the search index. Moreover, practical applications often imply search with conditions (filters) that are challenging to incorporate in these methods.

A paper by Yadav et al. (2022) is the most relevant for our research. The idea is to apply the matrix factorization to the query-item relevance matrix in order to represent it as a product of its submatrix containing only a few columns (relevances for random support items) and some other, explicitly

computable (see Section 3.2 for the details). Despite the simplicity of the idea and implementation, the authors have shown in detail the superiority of their algorithm over more complex approaches, such as dual encoders. Our work is motivated by this study: we show that the approach of Yadav et al. (2022) has theoretical guarantees and suggest several improvements that significantly boost the performance. As an enhancement of the original approach, in another paper, Yadav et al. (2023) proposed the idea of selecting support items for each query, but the time complexity of each query processing becomes linear in the number of elements, which makes the approach infeasible in most practical applications. On the contrary, our support items selection is performed in the pre-processing stage and does not increase the query time.

Finally, the most recent paper by Yadav et al. (2024) proposes an alternative solution to the problem. Their AXN algorithm dynamically learns the difference between the DE and CE predictions for each query independently (at the query time). Learning the difference is carried out through iteratively choosing a set of anchor (supporting) elements, calculating CE scores for them, and learning linear regression with embeddings of these elements as features and CE scores as targets. Some disadvantages of this method are that it requires previously trained embeddings of queries and documents and it has increased query processing time due to the need for iterative refining of all item relevances for each query. For completeness of our study, we use the AXN algorithm as one of our baselines.

## 3. Relevance-Based Embeddings

In general, the information (attributes) used to calculate the ground-truth item-to-query relevances $R(i, q)$ can be divided into three types: depending only on the query $q$, only on the item $i$, and on both of them. The key problem when constructing separate embeddings of items and queries in the common space (that can be used for searching for the nearest elements) is the inability to use information that depends on both query and item, which lowers the quality of relevance search.

To address the above-mentioned issue, we introduce relevance-based representations that describe each query by its relevance to a pre-selected set of items and, vice-versa, each item by its relevance to a pre-selected set of queries. We prove that, under certain conditions, any relevance function can be well estimated using only such individual vectors. This expressive power is a clear advantage of the proposed relevance-based embeddings over conventional dual-encoder models.

### 3.1. Preliminaries

Let $Q$ and $I$ be compact topological spaces of queries and items, respectively. Assume that we are given a relevance function $R : I \times Q \to \mathbb{R}$. In practice, $R$ is a complex relevance model that can be computationally expensive and rely on pairwise features.

Let $S_I \subset I$ and $S_Q \subset Q$ be some finite ordered sets of *support items* and *support queries*: $S_I = \{i_1, \ldots, i_m\}$, $S_Q = \{q_1, \ldots, q_n\}$. Let $R(i, S_Q)$ be a *relevance vector* of the item $i$ w.r.t. the set of support queries $S_Q$: $R(i, S_Q) = (R(i, q_1), \ldots, R(i, q_n))$. Similarly, $R(S_I, q)$ is a relevance vector of the query $q$ w.r.t. the set of support items $S_I$: $R(S_I, q) = (R(i_1, q), \ldots, R(i_m, q))$. By $R(S_I, S_Q)$ we denote a relevance matrix composed in a similar way.

### 3.2. CUR Approximation

Relevance vectors can be utilized in different ways and one of the possible approaches is to use the CUR decomposition (this approach was used by Yadav et al. (2022)). Using our notation, the relevance $R(i, q)$ is approximated as:

$$\tilde{R}(i, q) = \langle R(i, S_Q) \times \text{pinv}(R(S_I, S_Q)), R(S_I, q) \rangle, \quad (1)$$

where $\text{pinv}(X)$ is the pseudo-inverse matrix of $X$. As support queries, Yadav et al. (2022) take the set of train queries: $S_Q = Q_{train} \subset Q$.

Regarding the computational complexity, we note that the CUR approximation requires computing the matrix $R(I, S_Q)$ which takes $M \cdot |S_Q| = M \cdot |Q_{train}|$ CE calls, where $M$ is the total number of items in a database, which can be infeasible for large databases.

Our first theoretical result provides guarantees for the CUR approximation (1). Namely, we show that its regularized version approximates the true relevance arbitrarily closely in $L_2$. Formally, let $\text{pinv}_\lambda(A) = (A^T A + \lambda E)^{-1} A^T$ with $E$ being the identity matrix. Then, the regularized CUR approximation $\text{CUR}_\lambda$ is defined by (1) with $\text{pinv}_\lambda$ instead of $\text{pinv}$. For $\text{CUR}_\lambda$, we prove the following result (see Appendix A.3 for the proof).

**Theorem 3.1.** *Suppose that $I$ and $Q$ are equipped with the structure of a measure space and the integral of $R^4(i, q)$ over $I \times Q$ is finite. Then, the $\text{CUR}_\lambda$ approximation of $R$ can be chosen arbitrarily close to the true $R$ in $L_2(I \times Q)$ provided enough independently sampled support items and queries and sufficiently small $\lambda$.*

Our result gives theoretical support for the good performance of the CUR decomposition demonstrated by Yadav et al. (2022). In the next section, we show that stronger theoretical guarantees can be provided if we allow transforming query and item vectors into more powerful embeddings.

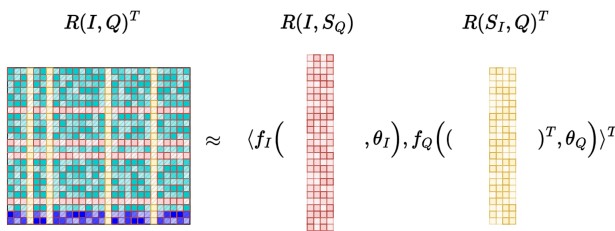

$R(I,Q)^T$      $R(I,S_Q)$      $R(S_I,Q)^T$

*Figure 1.* RBE visualization: support queries are red, support items are yellow; test queries are blue, and their relevance scores for the support items are used to approximate the remaining values.

### 3.3. Relevance-Based Embeddings

In this section, we propose extending the CUR-based approximation by allowing transformations of relevance vectors, e.g., with a neural architecture. With such embeddings, we prove a stronger result: that any continuous relevance function can be uniformly approximated.

We say that a function on $I$ is a *relevance-based embedding* if it has a representation of the form $e_I(i) = f_I(R(i, S_Q), \theta_I)$ where $S_Q$ is a set of support queries and $f_I$ is a neural architecture with parameters $\theta_I$ which parametrizes a mapping $\mathbb{R}^n \to \mathbb{R}^d$. Analogously, $e_Q(q) = f_Q(R(S_I, q), \theta_Q)$ is a relevance-based embedding of the query $q$.

The following theorem holds (the proof can be found in Appendix A.1 and the guarantees for the RBE approach on a sphere are discussed in Appendix A.2).

**Theorem 3.2.** *Let $I$ and $Q$ be compact topological spaces and $R : I \times Q \to \mathbb{R}$ be a continuous function. Then, $R$ can be uniformly approximated up to an arbitrarily small absolute error by a function $\tilde{R}(i, q)$:*

$$\tilde{R}(i,q) = \langle\, f_I(R(i, S_Q), \theta_I),\ f_Q(R(S_I, q), \theta_Q)\,\rangle, \quad (2)$$

*where $S_I \subset I$ and $S_Q \subset Q$ are some finite sets of support items and queries and $f_I$, $f_Q$ are neural architectures with the universal-approximation property (e.g., MLPs).*

This theorem shows that the true relevance function can be uniformly approximated with arbitrary precision by some functions of the relevance vectors. Note that compared to Theorem 3.1, Theorem 3.2 has weaker requirements and stronger convergence. Trainable embeddings allow us to prove uniform convergence for any continuous function $R$.

The proposed RBE framework is visualized in Figure 1. Note that the CUR approximation from Section 3.2 fits our framework with $\theta_I := \text{pinv}(R(S_I, S_Q))$ and $f_I(R(i, S_Q), \theta_I) := R(i, S_Q) \times \theta_I$, $f_Q(R(S_I, q), \theta_Q) := R(S_I, q)$. Extending the CUR approximation to arbitrary embeddings allows us to obtain even stronger approximation guarantees. Moreover, such trainable embeddings are more

flexible as they can adapt to a particular quality function one aims to optimize.

Regarding the computation complexity, the pre-processing step requires $M \cdot |S_Q| + N$ computations of CE, where $N$ is the size of the training set (the number of query-item pairs used for training,[1] this part is similar to the training of dual encoders). An advantage of RBE is that we may have a significantly smaller set of support queries $|S_Q| \ll |Q_{train}|$ while still utilizing the remaining queries for training the mappings $f_Q$ and $f_I$. Importantly, the number of CE computations can be further reduced to just $O(N)$ if we replace the transformation $f_I(R(i, S_Q), \theta_I)$ with trainable embeddings of items $\theta_I(i) \in R^d$ (see Section 3.5). The latter observation is a significant advantage of RBE over the CUR approximation on large databases.

To sum up, the main outcome of our analysis is that even though the relevance function $R(\cdot, \cdot)$ is arbitrary (and, in particular, can rely on pairwise features only available for query-item pairs), it can be well approximated by relevance-based embeddings of individual users and items.

### 3.4. Selecting Support Items

Let us revisit the statement of Theorem 3.2. The theorem states that there exist such sets $S_Q$, $S_I$ that the relevance function $R$ can be effectively approximated by separated embedders $f_I$, $f_Q$. In the related works (Morozov & Babenko, 2019; Yadav et al., 2022), the selection of $S_I$ is random, implicitly assuming that the elements are in some sense equivalent. However, for example, when building a recommendation service, the popularity of different objects has a strongly skewed distribution, which is why more information is known about a small set of highly popular items than about a large set of unpopular ones. Thus, it is natural to assume that the choice of support items may have a significant effect on performance. We investigate this direction and compare several approaches from simple heuristics to more complex ones optimizing the approximation quality.

Our **heuristic strategies** include:[2]

- **Random** Support items are chosen uniformly at random (Morozov & Babenko, 2019; Yadav et al., 2022). For better reproducibility, we also present the results of using the first $|S_I|$ items as support ones, assuming that the order of items is pseudorandom.

- **Popular** As mentioned above, a recommendation service usually has a small set of very popular elements that many users interact with. As a result, a lot of information can be collected from these interactions thus

---

[1] We assume that $N$ includes relevances of training queries to fixed support items.

[2] The approaches that require item representations can be applied to their relevance vectors.

making the popular elements more informative. Since it is not always possible to get popularity explicitly, we consider the following surrogate: choose the objects with the highest average relevance for the training set.

- **Cluster centers**   When it comes to the allocation of a representative subset of vectors, it is reasonable to consider the allocation of clusters. We consider various clustering algorithms and select the cluster centers as support elements. The number of clusters is set to the number of required support elements.

- **Most diverse**   This strategy is a greedy algorithm maximizing the minimum distance between the support elements. We first choose the element furthest from the center (by Euclidean distance) and then, at each step, an element is selected whose minimum distance to the current support elements is maximal.

$l_2$-**greedy approach**   Let us now discuss a more theoretically justified approach that we call $l_2$-greedy. It aims at selecting the support elements that allow for a better approximation of the relevance matrix $R(i, q)$. In this strategy, we greedily select items so that the MSE error of the CUR approximation (Mahoney & Drineas, 2009) is minimized for the train queries.[3]

We note that the CUR approximation replaces every item with a linear combination of support items so that the MSE between the true relevances and their CUR approximations on the train queries is minimized (see Appendix A.3). Our goal is to optimize the overall MSE for all items, which is:

$$\sum_i \| R(i, S_Q)^T - R(S_I, S_Q)^T$$
$$\times \text{pinv}(R(S_I, S_Q)^T) \times R(i, S_Q)^T \|_2^2. \quad (3)$$

We minimize this expression over all possible choices of $S_I$, $|S_I| = m$. We propose a greedy approach in which support items are selected one by one optimizing (3) at each step. The implementation details can be found in Appendix B.

**Computation complexity**   In their default implementation, support items selection strategies *popular*, *cluster centers*, *most diverse*, and $l_2$-*greedy* require computing the relevance scores of all items to support queries that can be done in $O(M \cdot n)$. If this is infeasible, one can use downsampling to reduce the number of candidates for support items. Our preliminary experiments (see Appendix F) show that even significant downsampling gives reasonable performance of the obtained support items. Moreover, for heuristic approaches,

---

[3]We use the CUR approximation since it optimizes MSE and is deterministically defined as in (1), while RBE requires fitting $f_I$ and $f_Q$ before estimating the approximation quality for each choice of the support elements.

instead of the relevance vectors, one can use cheaper embeddings (e.g., the original feature vectors) when they are available. Also, for *cluster centers*, there can be predefined clusters in the data. For instance, in recommendation services, it is typical that items are annotated with their categories that can be used as clusters without any additional cost.

### 3.5. Additional Practical Considerations

**Dynamic set of items**   We note that RBE can naturally handle scenarios where the set of items frequently changes. The embeddings $f_I(R(i, S_Q), \theta_I)$ can be easily calculated for the new items without the need of re-training the embedding model $f_I$ (similarly to feature-based dual encoders). On the other hand, if the set of items $I$ is finite or changes not so frequently (e.g., the set of movies currently available in a recommendation service), the transformation $f_I(R(i, S_Q), \theta_I)$ can be replaced with trainable embeddings $\theta_I(i) \in R^d$. In contrast to items, the query set $Q$ cannot be assumed to be finite: queries can be represented by texts of unlimited length or characterized by real-valued features.

**Improving approximation quality**   It is natural to assume the heavy ranker $R$ to be the most expensive part in terms of computational complexity. Thus, during the calculation of $\tilde{R}$, we are mainly limited by the sizes of the support sets $S_Q$ and $S_I$. In contrast, calculating $f_Q$, $f_I$, or the dot product between them is assumed to be significantly cheaper. Thus, $f_Q$ and $f_I$ may embed the relevance vectors in a higher-dimensional space. In particular, this may help to eliminate the disadvantages of the dot product, in comparison with other (Shevkunov & Prokhorenkova, 2021) ways of measuring the distances or similarities between objects.

While RBE has theoretical guarantees, they hold in the limit, when the sets of support queries and items are sufficiently large. In practice, to improve the approximation performance and reduce the number of CE calls per query, the mapping $f_Q$ (or $f_I$) could be extended by enriching it with the features of the original query (or item).

**Scalability**   The cost of the pre-processing and the ways to scale it are discussed in Section 3.3. At the inference, we need to compute the query representation, which requires $m$ relevance computations. Then, the inference is similar to dual encoders: the item representations are precomputed and placed in the approximate nearest neighbors index like HNSW (Malkov & Yashunin, 2020), which takes the embedding of the query as input. These $m$ additional computations are taken into account in our experiments: when comparing with dual encoders, we reduce the final re-ranking budget by $m$.

*Table 1.* Support item selection applied to the CUR approximation; HitRate(100) is reported (larger is better); we highlight the **first**, **second**, and **third** best results.

| Support items | Yugioh | P.Wrest. | StarTrek | Dr.Who | Military | RecGames2 | RecGames1 | RecMusic | QA.Small |
|---|---|---|---|---|---|---|---|---|---|
| random (i.e. AnnCUR) | 0.4724 | 0.4280 | 0.2287 | 0.1919 | 0.2455 | 0.6697 | 0.5842 | 0.1478 | **0.5828** |
| first 100 | 0.4845 | 0.4182 | 0.2489 | 0.1975 | 0.2599 | 0.6490 | 0.5609 | 0.1551 | 0.5491 |
| popular | 0.2429 | 0.3001 | 0.1154 | 0.1197 | 0.1907 | **0.7623** | **0.6695** | 0.1422 | 0.5536 |
| KMeans | 0.5083 | **0.4850** | 0.3226 | **0.2517** | **0.3042** | 0.7070 | 0.6184 | **0.1661** | 0.5578 |
| BisectingKMeans | 0.4825 | 0.4592 | 0.2839 | 0.2159 | 0.2752 | 0.7035 | 0.6213 | 0.1483 | 0.5394 |
| MiniBatchKMeans | 0.5077 | 0.4737 | 0.2912 | 0.2365 | **0.2826** | 0.7033 | 0.5981 | **0.1721** | 0.5529 |
| AgglomerativeClustering | **0.5105** | **0.4911** | **0.3264** | **0.2531** | 0.2448 | 0.7050 | **0.6265** | 0.1557 | 0.5666 |
| SpectralCoclustering | 0.4618 | 0.4443 | 0.2540 | 0.2076 | 0.2551 | 0.6998 | 0.6094 | 0.1594 | 0.5415 |
| SpectralBiclustering | 0.4654 | 0.4708 | 0.2628 | 0.1845 | 0.2533 | **0.7409** | 0.5972 | 0.1607 | 0.5358 |
| SpectralClusteringNN | 0.5087 | 0.4690 | 0.2742 | 0.2048 | 0.2507 | 0.6936 | 0.5740 | **0.2343** | 0.5741 |
| most diverse | **0.5333** | 0.4290 | **0.3325** | 0.2278 | 0.2483 | 0.6504 | 0.6182 | 0.1329 | **0.5925** |
| $l_2$-greedy | **0.5618** | **0.5119** | **0.3677** | **0.2960** | **0.3357** | **0.7197** | **0.6565** | 0.1478 | **0.6100** |

## 4. Experimental Setup

For all datasets that we use in this work, there is a heavy ranker that provides relevance. With this ranker, we build the table ($R : I \times Q \to \mathbb{R}$) of the relevance scores. The task is to find the most relevant items for a given query. We denote the predicted scores by $\tilde{R}$. The quality is evaluated as HitRate($P, T$), which equals[4]

$$\frac{1}{|Q_{test}|} \sum_{q_i \in Q_{test}} \frac{|\text{Best}_P(\tilde{R}, q_i) \cap \text{Best}_T(R, q_i)|}{|\text{Best}_T(R, q_i)|}.$$

Here $\text{Best}_K(\mathcal{R}, q) \subset I$ is defined as the set of $K$ items $i_1, \ldots, i_K$ with the highest relevances $\mathcal{R}(i_1, q), \ldots, \mathcal{R}(i_K, q)$ to a given query $q \in Q$; $Q_{test}$ is a set of test queries, $Q_{train} \bigsqcup Q_{test} = Q$. For all our experiments, $|Q_{test}| \approx 0.3|Q|$, $|S_I| = 100$, $S_Q = Q_{train}$. We denote HitRate($K$) := HitRate($K, K$).

### 4.1. Datasets

We evaluate our approach on the following datasets. The Zero-Shot Entity Linking (**ZESHEL**) dataset is constructed by Logeswaran et al. (2019) from Wikia. Following Yadav et al. (2022), we run the experiments on five domains from ZESHEL and use the cross-encoder trained by Yadav et al. (2022) as a heavy ranker $R$. To additionally cover the question-answering domain, we conducted experiments on the MS MARCO (Bajaj et al., 2016) dataset provided by Hugging Face. As a heavy ranker, we use *all-mpnet-base-v2* from the SentenceTransformers library (Reimers & Gurevych, 2019), which is trained, among other datasets, on the MS MARCO data.[5] We took ∼10K test queries and 0.8M passages corresponding to them (**QA**). For the experiments in Table 1, we used the smaller (**QA.Small**) version

with 82K passages (only the test passages).

We also collected datasets from Yandex Games and Yandex Music services providing recommendations of items to users. In both services, there is a heavy ranker $R$ that is the CatBoost gradient boosting model (Prokhorenkova et al., 2018) trained on a large set of external data and features, including those provided by multiple two-tower neural networks. There are also strong dual encoders trained on large data with rich features. A smaller subsample of this data was used to train RBE, since we believe that RBE is able to show good quality even on a significantly smaller size of the training data. For Yandex Games, there are two datasets **RecGames2** and **RecGames1** that differ by an objective used to train the heavy ranker. We refer to the Yandex Music dataset as **RecMusic**. The detailed description of the datasets is given in Appendix C.

Overall, in our experiments, we aim at showing that relevance vectors are powerful query and item representations and that they can be used in a variety of tasks when an arbitrary relevance function for query-item pairs is available. For this, we conduct experiments covering a variety of dimensions: tasks (entity linking, question answering, recommendations); relevance models (different heavy rankers including neural networks and gradient boosting); setups (both reproducible academic benchmarks and in-the-wild production systems with strongly tuned dual encoders). Thus, our setup allows us to demonstrate the usefulness and applicability of the proposed approach in diverse scenarios.

### 4.2. Baselines

As our main baseline, we consider the **AnnCUR** algorithm (Yadav et al., 2022) that approximates relevances with the CUR decomposition as discussed in Section 3.2. What is important for further discussion, a broad comparison of this method with different basic approaches, including various dual encoders, is carried out by Yadav et al. (2022). In most of our experiments, we rely on these results, comparing

---

[4]Our HitRate($k, k_r$) is equivalent to Top-$k$-Recall@$k_r$ from Yadav et al. (2022).

[5]Additional experiments with another CE can be found in Appendix E.2.

*Table 2.* Evaluating neural relevance-based embeddings; HitRate(100) is reported (larger is better).

| Model | Yugioh | P.Wrest. | StarTrek | Dr.Who | Military | RecGames2 | RecGames1 | RecMusic | QA |
|---|---|---|---|---|---|---|---|---|---|
| Popular | 0.0917 | 0.2410 | 0.0884 | 0.0821 | 0.1127 | 0.5077 | 0.2886 | 0.0142 | 0.0001 |
| AnnCUR | 0.4724 | 0.4280 | 0.2287 | 0.1919 | 0.2455 | 0.6697 | 0.5842 | 0.1478 | 0.5522 |
| AnnCUR+KMeans | 0.5083 | 0.4850 | 0.3226 | 0.2517 | 0.3042 | 0.7070 | 0.6184 | 0.1661 | 0.5027 |
| RBE+KMeans | 0.5431 | 0.4979 | 0.3399 | 0.2539 | 0.3019 | 0.7137 | 0.6300 | 0.3729 | 0.5505 |
| AnnCUR+$l_2$-greedy | 0.5618 | 0.5119 | 0.3677 | 0.2960 | **0.3357** | 0.7197 | 0.6565 | 0.1478 | 0.5700 |
| RBE+$l_2$-greedy | **0.5849** | **0.5249** | **0.3867** | **0.2992** | 0.3349 | **0.7234** | **0.6682** | **0.3964** | **0.6022** |

only with AnnCUR. However, we explicitly provide the comparison with **dual encoders** for the new recommendation datasets. In addition to dual encoders, we also compare with the **AXN** approach proposed by Yadav et al. (2024). This method iteratively adds candidate items and uses the heavy-ranker scores of the currently chosen candidates to improve the DE predictions for the remaining items. For better interpretability of the results, we also provide metrics for a baseline that always selects the most popular items (not to be confused with Table 1, where "popular" refers to selecting support items).

### 4.3. RBE Implementation

Following Theorem 3.2, we train lightweight neural networks $f_I$ and $f_Q$, which are significantly faster than the heavy ranker $R$, and independently transform the relevance vectors $R(i, S_Q), R(S_I, q)$ into embeddings. The training is performed by the Adam algorithm on sampled batches with a listwise loss function (see Appendix D), similar to the training of various DEs. As follows from Section 3.2, the CUR decomposition gives a reasonably good approximation of the relevance function. Hence, we split the RBE representation into the CUR representation and the trainable prediction of its error. In the experiments, such decomposition improves the convergence and training stability. Technical implementation details are placed in Appendix D.[6]

## 5. Experimental Results

**Support items selection** First, we compare various ways of choosing support elements described in Section 3.4. Here, all clustering algorithms are taken from the scikit-learn (Pedregosa et al., 2011) library, SpectralClusteringNN is a SpectralClustering with "nearest neighbors" affinity. The algorithms are used with their default parameters since even this simple setting already allows us to get significant improvements over the random selection. The results are shown in Table 1, where the best three results for each dataset are highlighted. Clearly, there is a significant superiority of

almost any approach based on clustering or diversity over random selection. The theoretically justified $l_2$-greedy algorithm is the clear winner, second and third places are taken by KMeans and AgglomerativeClustering. However, due to the significantly worse quality of AgglomerativeClustering on the Military dataset, KMeans will be used in further experiments. Another observation is that on RecGames, there is a clear superiority of the choice of popular items as the support ones. It is worth mentioning that for this dataset, the elements extracted by popularity are also quite stratified by their categories. However, this may not hold for other data (e.g., we do not observe a similar property for RecMusic).

**Neural relevance-based embeddings** Following Section 4.3, we also apply trainable relevance mappings $f_I(R(I, S_Q), \theta_I), f_Q(R(S_I, q), \theta_Q)$ to check whether this modification improves prediction quality in practice. The results are shown in Table 2. For better interpretability of the results, the quality of the constant output consisting of the most popular elements (those with the highest average relevance to the training queries) is also reported. In most cases, except for one dataset (Military), trainable relevance mappings improve the quality of the relevance search, and the improvements are obtained for both KMeans and $l_2$-greedy support elements selection. Improvement from the trainable mappings is most noticeable for the QA dataset. Thus, we see that with a simple lightweight transformation, one can get an increased quality on various datasets.

**Comparison with dual encoders** In this part, we compare RBE with dual encoders. For the RecGames and RecMusic datasets, we consider the embeddings produced by the dual encoder used in the production of the service (i.e., the one that is empirically proved to be the best in this task). For both DEs, the approximate sizes are provided in Table 3, confirming that we can call RBE a lightweight model.

*Table 3.* RBE and DE sizes (trainable parameters).

| RBE | DE, RecGames | DE, RecMusic |
|---|---|---|
| $\sim$ 50K | $\sim$ 300M | $\sim$ 700M |

To fairly compare the performance, we replace the relevance

---

[6]Code and experimental data are available at https://github.com/shevkunov/Relevance-Based-Embeddings-Lightweight-Candidate-Retrieval.

*Table 4.* DE vs RBE on RecGames1.

| $X$ | Dual Encoder | AXN$_{DE}$ | RBE+$l_2$-greedy |
|---|---|---|---|
| | HR$(X+100, X)$ | HR$(X+100, X)$ | HR$(X, X)$ |
| 100 | 0.7048 | **0.7065** | 0.6682 |
| 200 | 0.6803 | 0.6769 | **0.6955** |
| 300 | 0.6739 | 0.6740 | **0.7221** |
| 400 | 0.6739 | 0.6769 | **0.7406** |
| 500 | 0.6760 | 0.6820 | **0.7538** |
| 600 | 0.6792 | 0.6883 | **0.7639** |
| 700 | 0.6827 | 0.6958 | **0.7720** |
| | HR$(X+100, 100)$ | HR$(X+100, 100)$ | HR$(X, 100)$ |
| 100 | 0.7048 | **0.7065** | 0.6682 |
| 200 | 0.7977 | 0.7970 | **0.8359** |
| 300 | 0.8518 | 0.8538 | **0.9026** |
| 400 | 0.8855 | 0.8902 | **0.9342** |
| 500 | 0.9086 | 0.9153 | **0.9522** |
| 600 | 0.9258 | 0.9331 | **0.9632** |
| 700 | 0.9385 | 0.9465 | **0.9704** |

*Table 5.* DE vs RBE on RecMusic.

| $X$ | Dual Encoder | AXN$_{DE}$ | RBE+$l_2$-greedy |
|---|---|---|---|
| | HR$(X+100, X)$ | HR$(X+100, X)$ | HR$(X, X)$ |
| 100 | 0.3792 | 0.3843 | **0.3964** |
| 200 | 0.3198 | 0.3333 | **0.4471** |
| 300 | 0.2928 | 0.3015 | **0.4693** |
| 400 | 0.2784 | 0.2879 | **0.4833** |
| 500 | 0.2702 | 0.2839 | **0.4929** |
| 600 | 0.2661 | 0.2835 | **0.5008** |
| 700 | 0.2647 | 0.2836 | **0.5070** |
| | HR$(X+100, 100)$ | HR$(X+100, 100)$ | HR$(X, 100)$ |
| 100 | 0.3792 | 0.3843 | **0.3964** |
| 200 | 0.4228 | 0.4296 | **0.5435** |
| 300 | 0.4514 | 0.4557 | **0.6253** |
| 400 | 0.4738 | 0.4834 | **0.6819** |
| 500 | 0.4927 | 0.5020 | **0.7253** |
| 600 | 0.5091 | 0.5198 | **0.7599** |
| 700 | 0.5239 | 0.5351 | **0.7888** |

vectors in the RecGames and RecMusic experiments with embeddings obtained by the DE. The only difference between the pipelines for RBE and DE is that since we used $|S_I| = 100$ requests to a heavy ranker to obtain an RBE, we have to reduce the budget for RBE CE calls when calculating the top: in all the comparisons, DE uses $|S_I|$ more CE calls than RBE during re-ranking. We also evaluate the performance of AXN which uses the same number of CE calls for re-ranking as the dual encoder. We denote this method as AXN$_{DE}$ to specify that DE is used as a dual encoder within AXN. Thus, for DE and AXN, we use the metric HitRate$(X + |S_I|, Y)$ and for RBE — HitRate$(X, Y)$, which gives the former an advantage for small budget $X$.[7]

We conduct two sets of experiments. In the first one, the goal is to retrieve the best $X$ elements (for varying $X$) when the allowed number of CE calls changes accordingly and equals $X + 100$. In the second set of experiments, the goal is to retrieve the best 100 elements while the number of CE calls increases as $X + 100$. As can be seen from Table 4, for RecGames1, RBE is superior to both DE and AXN$_{DE}$ baselines starting from $X = 200$. As for RecMusic (Table 5), our algorithm is superior to the baselines starting from $X = 100$. Let us note that in the production service, the actual size of the top used to select candidates before ranking is typically large (significantly larger than 100).

A similar comparison with the dual encoder on the data from ZESHEL can be found in Yadav et al. (2022): it is shown that AnnCUR outperforms the dual encoder. Thus, on ZESHEL we compare only with AnnCUR.

**Other applications** Although the goal of RBE is to approximate the relevance function, relevance vectors can

also be considered good general representations. In Appendix E.3, we conduct an additional experiment on predicting the categories of items in the RecGames1 dataset and obtain that the relevance vectors are informative vector representations for this task. In particular, we show that categories are predicted better by vectors derived from relevance than by DE vectors and that selecting better support items for the relevance retrieval task also improves the quality of category prediction. This experiment demonstrates that the idea of representing elements using relevance vectors provided by a complex model has potential beyond relevance retrieval. See Appendix E.3 for the details.

## 6. Conclusion & Future Research

In this paper, we present the concept of relevance-based embeddings. We justify our approach theoretically and show its practical effectiveness on various academic and production datasets. We demonstrate that RBE allows for obtaining better quality compared with existing approaches while having a significantly smaller model size and lower training cost. Our theoretical and empirical analysis shows that the idea of using relevance embeddings is useful for various machine learning tasks.

**Limitations and directions for future research** The proposed approach relies on the availability of the strong final ranking model, which is a key ingredient for any recommendation service. If the heavy ranker is not sufficiently reliable, then the candidates provided by our approach, while being better according to the relevance model, may lead to worse user experience.

Promising directions for future research include a theoretical analysis of the convergence rate, a deeper investigation

---

[7]In the tables, 'HitRate' is shortened to 'HR'.

of support element selection strategies as well as applying the proposed RBE to other algorithms, e.g., based on using a heavy ranker during the nearest neighbor search (Morozov & Babenko, 2019) or adaptive nearest neighbor search (Yadav et al., 2024). Regarding the latter approach, we note that AXN can be naturally combined with any query-item representations, e.g., RBE. The obtained $AXN_{RBE}$ model should, in principle, perform at least as well as RBE due to the presence of the regularization that balances between the query-item representation and the proposed modification.

## Acknowledgements

We thank Daniil Burlakov for providing data for the Yandex Music dataset.

## Impact Statement

This paper presents work whose goal is to advance the field of Machine Learning. There are many potential societal consequences of our work, none of which we feel must be specifically highlighted here.

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

# A. Theoretical Analysis

## A.1. Proof of Theorem 3.2

**Queries as functions on items and vice versa**   Each query $q$ defines a function $r_q$ on items: $r_q(i) = R(i, q)$. Let us call two queries $q$ and $q'$ *R-equivalent* if $r_q = r_{q'}$ and write $q \sim_R q'$ to denote this relation. $R$-equivalent queries are interchangeable when it comes to measuring their relevance to any item. Let $Q_R$ be the set of $R$-equivalence classes. $Q_R$ may be considered as an image of $Q$ in $C(I)$ under the mapping $R_Q$ which maps query $q$ to $r_q$. This point of view suggests a natural metric $d_{Q_R}$ on $Q_R$ induced by the uniform norm on $C(I)$: $d_{Q_R}(q, q') = \|r_q - r_{q'}\| = \sup_{i \in I} |R(i, q) - R(i, q')|$.

Now let us note that the map $R_Q : Q \to C(I)$ is continuous since $R$ is continuous and $I$ is compact. Therefore, $Q_R$ is compact as an image of a compact space $Q$ under a continuous mapping. And there is an (injective) embedding of $C(Q_R)$ to $C(Q)$ under which the function $f \in C(Q_R)$ goes to $f \circ R_Q \in C(Q)$. Simply speaking, a continuous function on the equivalence classes of queries is also a continuous function on the queries themselves.

Analogously, we define:

$$R_I : I \to C(Q), \quad R_I(i) = r_i, \quad r_i(q) = R(i, q), \quad R_I(I) = I_R,$$
$$d_{I_R}(i, i') = \|r_i - r_{i'}\| = \sup_{q \in Q} |R(i, q) - R(i', q)|.$$

For convenience, we will identify functions in $C(I_R)$ and $C(Q_R)$ with functions in $C(I_R \times Q_R)$ which are independent of one of their arguments. The relationships mentioned above and similar ones are shown in the diagram below (hooked arrows represent injective mappings, arrows with two heads stand for surjective ones):

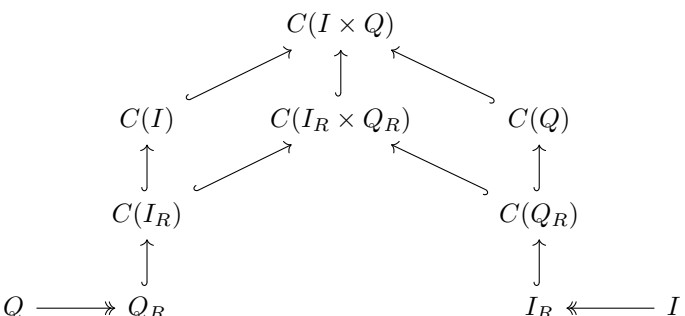

Now, let us make several observations.

**Claim 1.** $r_i \in C(Q_R)$ *and, similarly,* $r_q \in C(I_R)$.

*Proof.* We note that $\|r_i(q) - r_i(q')\| = \|r_q(i) - r_{q'}(i)\| \leq \|r_q - r_{q'}\| = d_{Q_R}(q, q')$. So, $r_i(q) - r_i(q') = 0$ if $r_q = r_{q'}$ and the value $r_i(q)$ does not change if a query is replaced with an equivalent one. It means that $r_i$ is a correctly defined function on the classes of equivalent queries, i.e., on $Q_R$. And finally the same inequality $\|r_i(q) - r_i(q')\| \leq d_{Q_R}(q, q')$ implies that the function $r_i$ is 1-Lipschitz with respect to the metric $d_{Q_R}$. $\square$

**Claim 2.** $R \in C(I_R, Q_R)$.

*Proof.* We have $|R(i, q) - R(i, q')| = |r_q(i) - r_{q'}(i)| \leq \|r_q - r_{q'}\| = d_{Q_R}(q, q')$ and $|R(i, q) - R(i', q)| \leq d_{I_R}(i, i')$. It follows that the value $R(i, q)$ does not change after replacement of a query-item pair $(i, q)$ with some equivalent pair $(i', q')$. So, $R$ can be considered as a function on $I_R \times Q_R$. And by the same inequality $R$ is 1-Lipschitz with respect to the metric $d_R((i, q), (i', q')) = d_{I_R}(i, i') + d_{Q_R}(q, q')$ and hence is continuous. $\square$

**Stone-Weierstrass theorem**   Let us call all functions $r_q$ and $r_i$ *elementary*. Consider the family of all elementary functions $\mathcal{F} = \{r_q | q \in Q\} \cup \{r_i | i \in I\} \subset C(I_R \times Q_R)$.

**Claim 3.** *The family $\mathcal{F}$ separates points in $I_R \times Q_R$, i.e., for each two different points $x, y \in I_R \times Q_R$, there is a function $f \in \mathcal{F}$ such that $f(x) \neq f(y)$.*

*Proof.* Indeed, let $(i_1, q_1)$ and $(i_2, q_2)$ be any two different points in $I_R \times Q_R$. Then $i_1 \neq i_2$ or $q_1 \neq q_2$. Without loss of generality, we can assume that $i_1 \neq i_2$ (they are unequal as points in $I_R$). So, $r_{i_1}$ and $r_{i_2}$ are different functions on $Q_R$ and there exists $q \in Q_R$ such that $r_{i_1}(q) \neq r_{i_2}(q) \Leftrightarrow R(i_1, q) \neq R(i_2, q) \Leftrightarrow r_q(i_1) \neq r_q(i_2) \Leftrightarrow r_q((i_1, q_1)) \neq r_q((i_2, q_2))$. Thus, we found a function $(r_q)$ from our family that separates the two points. $\square$

Next, consider the algebra of functions $\mathbb{R}[\mathcal{F}]$ generated by the family $\mathcal{F}$. This algebra consists of all polynomial combinations of functions in $\mathcal{F}$. More formally, each element of $\mathbb{R}[\mathcal{F}]$ has a representation of the form:

$$\mathbb{R}[\mathcal{F}] \ni f = \sum_{k=1}^{d} c_k \cdot r_{i_{k,1}} \cdot \ldots \cdot r_{i_{k,a_k}} \cdot r_{q_{k,1}} \cdot \ldots \cdot r_{q_{k,b_k}}.$$

In other words, there are $d$ sets $S^1, \ldots, S^d$ of queries and items such that:

$$S^k = S_I^k \cup S_Q^k, \quad S_I^k = \{i_{k,1}, \ldots, i_{k,a_k}\} \subset I, \quad S_Q^k = \{q_{k,1}, \ldots, q_{k,b_k}\} \subset Q.$$

$$f = \sum_{k=1}^{d} c_k \cdot \left( \prod_{i \in S_I^k} r_i \right) \cdot \left( \prod_{q \in S_Q^k} r_q \right). \tag{4}$$

Products of the form $\prod_{i \in S_I^k} r_i$ may be empty and in this case the product equals 1. So, $\mathbb{R}[\mathcal{F}]$ contains constant functions and separates points of $I_R \times Q_R$ (because it contains $\mathcal{F}$). Hence, by the Stone-Weierstrass theorem, the algebra $\mathbb{R}[\mathcal{F}]$ is dense in $C(I_R \times Q_R)$. In particular, the function $R$ can be approximated by an element of $\mathbb{R}[\mathcal{F}]$ up to an arbitrarily small absolute error.

**Represent polynomials in $\mathbb{R}[\mathcal{F}]$ as products of query and item embeddings** Consider an arbitrary function $f \in \mathbb{R}[\mathcal{F}]$ and its representation of the form (4). Denote the products $\prod_{i \in S_I^k} r_i(q)$ and $\prod_{q \in S_Q^k} r_q(i)$ by $\pi_{S_I^k}(q)$ and $\pi_{S_Q^k}(i)$ respectively. Consider two $d$-dimensional vectors:

$$e(q) = \left( c_1 \cdot \pi_{S_I^1}(q), \ldots, c_d \cdot \pi_{S_I^d}(q) \right),$$
$$e(i) = \left( \pi_{S_Q^1}(i), \ldots, \pi_{S_Q^d}(i) \right).$$

Then, $f(i, q) = \langle e(i), e(q) \rangle$. Let $S_I = \cup_{k=1}^{d} S_I^k$ and $S_Q = \cup_{k=1}^{d} S_Q^k$. Then $e(q)$ is a continuous (more specifically, polynomial) function of the vector $R(S_I, q)$ and $e(i)$ is a continuous function of the vector $R(i, S_Q)$. So, by the universality theorem for MLPs (Cybenko, 1989; Leshno et al., 1993), the vector $e(i)$ can be approximated up to arbitrarily small absolute error in the form $f_I(R(i, S_Q), \theta_I)$ where $f_I(\cdot, \theta_I)$ — a rich enough MLP architecture. Similarly, $e(q)$ can be approximated by $f_Q(R(S_I, q), \theta_Q)$. Hence, $\langle f_I(R(i, S_Q), \theta_I), f_Q(R(S_I, q), \theta_Q) \rangle$ approximates $f(i, q)$. Finally, we can consider $f \in \mathbb{R}[\mathcal{F}]$ such that $\|f - R\| < \frac{\varepsilon}{2}$ and then find such $\theta_I$ and $\theta_Q$ that $\|f - \langle f_I(R(i, S_Q), \theta_I), f_Q(R(S_I, q), \theta_Q) \rangle\| < \frac{\varepsilon}{2}$. These parameters will give us a desired $\varepsilon$-approximation of $R$ in a form of the product of relevance-based embeddings.

### A.2. RBE on a Sphere

The corollary below shows that the retrieval of the most $R$-relevant items with tolerance to $\varepsilon$-sized relevance loss can be reduced to the standard nearest neighbor search *on a sphere*.

**Corollary A.1.** *For each $\varepsilon > 0$, there is a multiplier $a \in \mathbb{R}$ such that $a\tilde{R}$ is an $\varepsilon$-approximation of $R$ and $\tilde{R}$ uses embeddings scaled to the unit sphere.*

*Proof.* Let us take some $\frac{\varepsilon}{2}$-approximation of $R$ of the form

$$R(i, q) \approx \langle e_Q(q), e_I(i) \rangle = \langle f_Q(R(S_I, q), \theta_Q), f_I(R(i, S_Q), \theta_I) \rangle$$

via the relevance-based embeddings $e_Q(q)$ and $e_I(i)$ of dimension $d$. Take a constant $C$ such that $\|e_I(i)\| < C$ and $\|e_Q(q)\| < C$ for all $q \in Q$ and $i \in I$ and consider the following vector embeddings of dimension $d + 2$:

$$\tilde{e}_I(i) = \left( \frac{1}{C} e_I(i), \sqrt{1 - \|\frac{1}{C} e_I(i)\|^2}, 0 \right),$$

$$\tilde{e}_Q(q) = \left( \frac{1}{C} e_q(q), 0, \sqrt{1 - \|\frac{1}{C} e_q(q)\|^2} \right).$$

Note that $\langle e_I(i), e_Q(q) \rangle = C^2 \cdot \langle \tilde{e}_I(i), \tilde{e}_q(q) \rangle$ and $\tilde{e}_I(i)$ and $\tilde{e}_q(q)$ lie on the $(d+1)$-dimensional unit sphere $S^{d+1} \subset \mathbb{R}^{d+2}$. We can take new powerful enough architectures $\tilde{f}_I$ and $\tilde{f}_Q$ with outputs normalized to unit sphere in $\mathbb{R}^{d+2}$ and fit for them parameters $\tilde{\theta}_I$ and $\tilde{\theta}_Q$ such that $\tilde{f}_I(R(i, S_Q), \tilde{\theta}_I) \approx \tilde{e}_I(i)$ and $\tilde{f}_Q(R(S_I, q), \tilde{\theta}_Q) \approx \tilde{e}_Q(q)$ and $\tilde{R}(i, q) = \langle \tilde{f}_I(R(i, S_Q), \tilde{\theta}_I), \tilde{f}_Q(R(S_I, q), \tilde{\theta}_Q) \rangle \approx \langle \tilde{e}_I(i), \tilde{e}_Q(q) \rangle$. More specifically, take $\tilde{\theta}_I$ and $\tilde{\theta}_Q$ such that:

$$|\langle \tilde{e}_I(i), \tilde{e}_Q(q) \rangle - \tilde{R}(i, q)| < \frac{\varepsilon}{2C^2} \Rightarrow |\langle e_I(i), e_Q(q) \rangle - C^2 \tilde{R}(i, q)| < \frac{\varepsilon}{2}. \tag{5}$$

Given that $|R(i, q) - \langle e_I(i), e_Q(q) \rangle| < \frac{\varepsilon}{2}$, it yields $|R - a\tilde{R}| < \varepsilon$. Which means that the statement of the corollary is satisfied with $a = C^2$.

$\square$

## A.3. Proof of Theorem 3.1

**Preliminaries**  Let us start with some notation. We assume that the spaces of items and queries $I$ and $Q$ are equipped with the structure of a measure space and probabilistic measures $\mu_I$ and $\mu_Q$, respectively. For any item $i$, by $r_i$ we denote a function on $Q$ such that $r_i(q) = R(i, q)$. By $L_2(I) = L_2(I, \mu_I)$ we denote a Hilbert space of measurable functions whose square is integrable. We also denote $\mathbb{E}_i f(i) = \int_I f d\mu_I$ for $f \in L_2(I)$.

Recall that with the CUR decomposition, the relevance $R(i, q)$ is approximated as:

$$\tilde{R}(i, q) = \langle R(i, S_Q) \times \mathrm{pinv}(R(S_I, S_Q)), \quad R(S_I, q) \rangle = \sum_{t=1}^{|S_I|} c_t \cdot r_{i_t}(q) \tag{6}$$

with some coefficients $c_t$ that are defined as:

$$\mathbf{c} = (c_1, \ldots, c_{|S_I|}) = [R(i, S_Q) \times \mathrm{pinv}(R(S_I, S_Q))]^T = \mathrm{pinv}(R(S_I, S_Q)^T) \times R(i, S_Q)^T. \tag{7}$$

The last equality holds because $\mathrm{pinv}(A)^T = \mathrm{pinv}(A^T)$.

For any vector $v \in \mathbb{R}^{|S_Q|}$:
$$\mathrm{pinv}(R(S_I, S_Q)^T) v \in \arg\min_{x \in \mathbb{R}^{|S_I|}} \|R(S_I, S_Q)^T x - v\|_2^2.$$

Thus, the coefficients $\mathbf{c}$ minimize the MSE between $R(i, S_Q)^T$ and $R(S_I, S_Q)^T \mathbf{c}$, i.e., between the relevances of queries from $S_Q$ to the item $i$ and $\sum_{t=1}^{|S_I|} c_t \cdot r_{i_t}$. Thus, calculating the item embedding $\mathbf{c}$ is merely solving a linear regression.

Then, $\mathrm{CUR}_\lambda$ is the analog of the CUR approximation that uses $l_2$ regularization with coefficient $\lambda$ while solving these multiple linear regression problems. Formally, recall that we define

$$\mathrm{pinv}_\lambda(A) = (A^T A + \lambda E)^{-1} A^T$$

with $E$ being the identity matrix of a proper size. Then, $\mathrm{CUR}_\lambda$ uses $\mathrm{pinv}_\lambda$ instead of $\mathrm{pinv}$ in (6).

Now, we are ready to prove the theorem. We do it in the following steps.

**Step 1**  We note that the function $R$ can be represented in the form of a no more than countable sum:

$$R(i, q) = \sum_{k=0}^{K} \lambda_k f_k(i) h_k(q),$$

where $K \in \mathbb{N} \cup \infty$ and $\{f_0, f_1, \ldots\}$, $\{h_0, h_1, \ldots\}$ are orthonormal sets of vectors in $L_2(I)$ and $L_2(Q)$, respectively. Equality holds almost everywhere on $I \times Q$ with respect to the measure $\mu_I \times \mu_Q$. Note that this statement is a generalization of finite dimensional PCA. Without loss of generality, we can assume that $\lambda_k \geqslant \lambda_{k+1}$. Then, the approximation $R(i, q) \approx \sum_{k=1}^n \lambda_k f_k(i) h_k(q)$ is a function on $I \times Q$ of rank $n$ closest to $R$ in $L_2(I \times Q)$ (like in ordinary PCA).

To prove this, we consider an operator $A : L_2(I) \to L_2(Q)$, $A(f)(q) = E_{i \sim \mu_I} f(i) R(i, q)$. $A$ is a Hilbert-Schmidt integral operator and hence it is compact. So, $A^* A$ is compact self-adjoint positive semi-definite operator in $L_2(I)$. By the spectral theorem for compact operators, $A^* A(v) = \sum_k \lambda_k^2 f_k \langle f_k, v \rangle$, where $\{f_1, \ldots\}$ is at most countable orthonormal set. Taking $h_k = \frac{1}{\lambda_k} A(f_k)$ completes the construction.

**Step 2**   Let $\{i_1, i_2, \ldots\}$ be an infinite sequence of independently sampled items. Then, with probability one (with respect to sampling of $\{i_1, i_2, \ldots\}$), there is $N$ large enough that:

$$\mathrm{E}_i \left( \min_{i' \in \{i_1, \ldots, i_N\}} \|r_i - r_{i'}\|_2^2 \right) < \varepsilon.$$

In other words, for every item $i$, there is some replacement $r(i) \in \{i_1, \ldots, i_N\}$ such that $\mathbb{E}_{i,q} |R(i, q) - R(r(i), q)|^2 < \varepsilon$. To prove that, we need resolve some technical issues.

**Substep 2.1**   Let $H$ be a countably dimensional (and hence separable) subspace of $L_2(Q)$ generated by $\{h_1, h_2, \ldots\}$. Then, $r_i \in H$ with probability one (with respect to the measure $\mu_I$).

We have $r_i(q) - \sum_{k=0}^K \lambda_k f_k(i) h_k(q) = 0$ almost everywhere on $I \times Q$, so for almost every $i$: $r_i(q) = \sum_{k=0}^K \lambda_k f_k(i) h_k(q)$ for almost every $q$. Also, $\|r_i\|_2 < \infty$ for almost every $i$ (otherwise $\|R\|_2$ could not be finite). So, for $i$ such that both $\|r_i\|_2 < \infty$ and $r_i(q) = \sum_{k=0}^K \lambda_k f_k(i) h_k(q)$ holds that $r_i \in H$. Thus, further we can assume that $r_i \in H$ (ignoring the set of "bad" items of measure 0).

**Substep 2.2**   The mapping $e : i \to r_i \in H$ is measurable with respect to the Borel $\sigma$-algebra on $H$.

Let $B_\delta(h)$ be a ball of radius $\delta$ around $h \in H$. Then, it is sufficient to show that for every $h \in H$ and $\delta > 0$ the set of items $e^{-1}(B_\delta(h))$ is measurable in $I$. Let $f(i) = \int_q |R(i, q) - h(q)|^2 d\mu_Q$. The function $|R(i, q) - h(q)|^2$ is integrable over $I \times Q$. So by the Fubini's theorem, $f$ is integrable over $I$. In particular, the set $\{i | f(i) < \delta^2\} = e^{-1}(B_\delta(h))$ is measurable.

**Substep 2.3**   $\forall \varepsilon > 0$ for almost every $i \in I$: $P_{i' \sim \mu_I}(\|r_i - r_{i'}\| < \varepsilon) > 0$.

Consider the Borel measure $e_*(\mu_I)$ on $H$ (the image of $\mu_I$ under the mapping $e$: $e_*(\mu_I)(A) = \mu_I(e^{-1}(A))$ or, informally speaking, the distribution of all $r_i$ in $H$). Let $X \subset H$ be a countable dense subset in $H$. Consider the set $X_{\varepsilon/2} = \{x \in X | e_*(\mu_I)(B_{\varepsilon/2}(x)) = 0\}$ and $Y = \cup_{x \in X_{\varepsilon/2}} B_{\varepsilon/2}(x)$. $Y$ is the union of a countable set of balls of measure zero, so $e_*(\mu_I)(Y) = 0$. For $h$ such that $e_*(\mu_I)(B_\varepsilon(h)) = 0$ there is $x \in X$ such that $\|h - x\| < \varepsilon/2$. $B_{\varepsilon/2}(x) \subset B_\varepsilon(h)$, hence $e_*(\mu_I)(B_{\varepsilon/2}(x)) = 0$, hence $x \in X_{\varepsilon/2}$, hence $h \in B_{\varepsilon/2}(x) \subset Y$. So, $\{h \in H | P_i(\|r_i - h\| < \varepsilon)\} \subset Y$ and the measure of this set is zero which yields the required statement.

**Substep 2.4**   Take our infinite sequence of independently sampled items $\{i_1, i_2, \ldots\}$. Consider the sequence of functions:

$$f_n(i, i_1, \ldots, i_n) = \min_{i' \in \{i_1, \ldots, i_n\}} \|r_i - r_{i'}\|_2^2.$$

We know that for every $i$: $P_{i'}(\|r_i - r_{i'}\|_2^2 < \varepsilon^2) > 0$. So with probability one some item $i_k$ will fall into $B_\varepsilon(r_i)$ and $\forall n \geqslant k : f_n(i, i_1, \ldots, i_n) < \varepsilon^2$. It follows that $f_n \to 0$ almost everywhere (on some large measure space where all independent variables $i, i_1, i_2, \ldots$ are defined). The sequence $f_n$ is bounded by the integrable function $f_1(i, i_1) = \|r_i - r_{i_1}\|_2^2$.

It follows that for almost every item $i$ and an infinite sequence $\{i_1, \ldots\}$ of items independently sampled from $\mu_I$,

$$\lim_{k \to \infty} \left( \min_{i' \in \{0\} \cup \{i_1, \ldots, i_k\}} \|r_i - r_{i'}\|_2^2 \right) = 0.$$

The expression inside the $\lim$ is bounded by $\|r_i\|_2^2$. Hence, by the Lebesgue's dominated convergence theorem, its mean over $i \in I$ tends to zero.

**Step 3**   Take $N$ large enough that

$$E_i \left( \min_{i' \in \{0\} \cup \{i_1, \ldots, i_N\}} \|r_i - r_{i'}\|_2^2 \right) < \varepsilon.$$

Take $\{i_1, \ldots, i_N\}$ as our support items. Let $I_N = \mathrm{span}(r_{i_1}, \ldots, r_{i_N})$ be a linear span of the first $N$ random items in $L_2(Q)$. Then, for each choice of the regularization coefficient $\lambda$:

$$E_i \min_{c_1, \ldots, c_N} \left( \lambda \sum_{k=1}^N c_k^2 + \|r_i - \sum_{k=1}^N c_k r_{i_k}\|_2^2 \right) < \lambda + \varepsilon.$$

For $v \in L_2(Q)$, let $proj_\lambda(v, I_N)$ be a linear combination $c_1 r_{i_1} + \ldots + c_N r_{i_N}$ that minimizes $\lambda \sum_{k=1}^N c_k^2 + \|v - \sum_{k=1}^N c_k r_{i_k}\|_2^2$. So, $E_i \|r_i - proj_\lambda(r_i, I_N)\|_2^2 < \lambda + \varepsilon$. Take $\lambda = \varepsilon$.

**Step 4**   We know that $proj_\varepsilon(r_i, I_N)$ is close to $r_i$ (the average $l_2$ distance is at most $\sqrt{2\varepsilon}$). Next, we will prove that the linear combination of the support items by which the regularized CUR approximates $r_i$ is close to $proj_\varepsilon(r_i, I_N)$ (on average).

Let $\{q_1, q_2, \ldots\}$ be a sequence of random queries independently sampled from $\mu_Q$. Let $Q_m$ be a set of the first $m$ queries. Then, for any pair of items $i, i' \in I$ let $\langle i, i' \rangle_m = \frac{1}{m} \sum_{k=1}^m R(i, q_k) \cdot R(i', q_k)$. In other words, $\langle i, i' \rangle_m$ is the Monte-Carlo estimate of the product of $r_i$ and $r_{i'}$ in $L_2(Q)$. Consider $N \times m$ matrix $\hat{R}$ such that $\hat{R}_{a,b} = R(i_a, q_b)$. The matrix $\hat{G} = \hat{R}\hat{R}^T$ consists of pairwise estimates of products of support items $\hat{G}_{kl} = \langle i_k, i_l \rangle_m$. As $m$ tends to infinity, the matrix $\hat{G}$ converges to the matrix $G$ of the exact scalar products of support items in the space $L_2(Q)$: $G_{kl} = E_q(r_{i_k}(q) r_{i_l}(q))$.

For each item $i$, let $\langle i, I_N \rangle_m$ be a vector $(\langle i, i_1 \rangle_m, \ldots, \langle i, i_N \rangle_m)$ and $\langle i, I_N \rangle = (\langle i, i_1 \rangle, \ldots, \langle i, i_N \rangle) = E_q \langle i, I_N \rangle_m$. Each component of $\langle i, I_N \rangle_m$ is the average of $m$ random variables of the form $r_i(q) r_{i_k}(q)$ which are conditionally mutually independent given $i$ and $I_N$. Let us estimate the mean squared deviation of $\langle i, I_N \rangle_m$ from $\langle i, I_N \rangle$ (with the fixed set of support items $I_N$):

$$E_{i,q_1,\ldots,q_m} \|\langle i, I_N \rangle_m - \langle i, I_N \rangle\|_2^2 = \frac{1}{m} \sum_{k=1}^N E_{i,q}(\langle r_i, r_{i_k} \rangle - r_i(q) r_{i_k}(q))^2$$

$$\leqslant \frac{1}{m} \sum_{k=1}^N E_{i,q} r_i^2(q) r_{i_k}^2(q)) \leqslant \frac{1}{m} \sum_{k=1}^N E_i \|r_i^2\| \|r_{i_k}^2\|$$

$$= \frac{1}{m} E_i \|r_i^2\| \sum_{k=1}^N \|r_{i_k}^2\| \leqslant \frac{1}{m} \|R^2\| \sum_{k=1}^N \|r_{i_k}^2\|.$$

So, the mean squared deviation of $\langle i, I_N \rangle_m$ from the vector $\langle i, I_N \rangle$ tends to zero as $m \to \infty$.

Finally, let $A$ be an operator that maps the coefficients $c_1, \ldots, c_N$ to linear combinations of support items $c_1 r_{i_1} + \ldots + c_N r_{i_N} \in L_2(Q)$. Note that the operator norm $\|A\|$ of A is finite as it is a finite rank operator (so $\|Av\| \leqslant \|v\| \|A\|$).

We have

$$proj_\varepsilon(r_i, I_N) = A(G + \varepsilon E_N)^{-1} \langle i, I_N \rangle,$$

where $E_N$ is $N \times N$ identity matrix. While the regularized CUR approximation of $r_i$ is:

$$CUR_\varepsilon(r_i, I_N) = A(\hat{G} + \varepsilon E_N)^{-1} \langle i, I_N \rangle_m.$$

Let $\delta_m$ be the vector $\langle i, I_N \rangle_m - \langle i, I_N \rangle$ and $\delta G_m^{-1}$ be $(\hat{G} + \varepsilon E_N)^{-1} - (G + \varepsilon E_N)^{-1}$. Then, the norm of the difference between $CUR_\varepsilon(r_i, I_N)$ and $proj_\varepsilon(r_i, I_N)$ can be estimated as:

$$\|CUR_\varepsilon(r_i, I_N) - proj_\varepsilon(r_i, I_N)\|$$

$$= \|A((G + \varepsilon E_N)^{-1} + \delta G_m^{-1})(\langle i, I_N \rangle + \delta_m) - A(G + \varepsilon E_N)^{-1} \langle i, I_N \rangle\|$$

$$= \|A(\delta G_m^{-1} \langle i, I_N \rangle + \delta G_m^{-1} \delta_m + (G + \varepsilon E_N)^{-1} \delta_m)\|$$

$$\leqslant \|A\|(\|\delta G_m^{-1}\| \|\langle i, I_N \rangle\| + \|\delta G_m^{-1}\| \|\delta_m\| + \|(G + \varepsilon E_N)^{-1}\| \|\delta_m\|)$$

$$\leqslant \|A\| \cdot (\|\delta_m\| \frac{1}{\varepsilon} + \|\delta G_m^{-1}\| \|\langle i, I_N \rangle\| + \|\delta_m\| \|\delta G_m^{-1}\|),$$

where we use $\|(G + \varepsilon E_N)^{-1}\| \leqslant \frac{1}{\varepsilon}$.

We can take $m$ large enough that the expectation of the square of that difference is arbitrarily small, say less than $\varepsilon$. Then,

$$\mathrm{E}_i \|r_i - CUR_\varepsilon(i)\|^2 \leqslant 2\mathrm{E}_i \|r_i - proj_\varepsilon(r_i, I_N)\|^2 + 2\mathrm{E}_i \|proj_\varepsilon(r_i, I_N) - CUR_\varepsilon(i)\|^2 \leqslant 6\varepsilon. \qquad (8)$$

## B. Greedy Selection of Support Items

Let us denote $X := (I, S_Q)$, which is an $M \times n$ matrix ($M$ is the total number of items). Then, our optimization problem can be formulated as follows.

We are given an $M \times n$ matrix $X$ of real numbers and let $x_i^T$, $i = 1, \ldots, M$, be the rows of $X$. Choose $m$ rows in such a way that the sum of squared distances from each row of the matrix to the space generated by the chosen rows would be minimal. In other words, find a subset of indices $S = \{i_1, \ldots, i_m\} \subset \{1, \ldots, M\}$ which minimizes following expression:

$$\sum_{i=1}^{M} \|x_i - \pi(x_i, \mathrm{span}(x_{i_1}, \ldots, x_{i_m}))\|_2^2 = \sum_{i=1}^{M} \|x_i - X_S^T \mathrm{pinv}(X_S^T) x_i\|_2^2,$$

where $\pi(v, V)$ is orthogonal projection of vector $v$ to subspace $V$, $X_S$ is an $m \times n$ matrix consisting of rows with indices from $S$. This problem corresponds to the CUR-decomposition of $X$ with $m$ rows and all $n$ columns.

A straightforward way is to choose items greedily. Suppose we have already chosen items $i_1, \ldots, i_t$. Then, we choose an item $i_{t+1}$ so that

$$\sum_{i=1}^{M} \|x_i - \pi(x_i, \mathrm{span}(x_{i_1}, \ldots, x_{i_{t+1}}))\|_2^2$$

is minimal.

Let us discuss how to choose $x_{i_{t+1}}$. Let $\Delta^t$ be the $M \times n$ matrix of our current approximation errors: $\Delta_i^t = x_i - \pi(x_i, \mathrm{span}(x_{i_1}, \ldots, x_{i_t}))$, $\Delta^0 = X$. Note that $\mathrm{span}(x_{i_1}, \ldots, x_{i_t}, x_i) = \mathrm{span}(x_{i_1}, \ldots, x_{i_t}, \Delta_i^t / \|\Delta_i^t\|_2)$, so for the purpose of evaluation our objective we can replace $x_i$ with $o_i^t = \Delta_i^t / \|\Delta_i^t\|_2$. When we add $x_i$ to the support set, the squared error on $x_j$ reduces by $\langle x_j, o_i^t \rangle^2$ and $\Delta_j^t$ becomes $\Delta_j^t - \langle x_j, o_i^t \rangle o_i^t$. It can be seen by considering the orthonormal basis of $\mathbb{R}^m$, the first $t$ elements of which generate $\mathrm{span}(x_{i_1}, \ldots, x_{i_t})$ and $(t+1)$-th is $o_i^t$. Adding $o_i^t$ to the support set will set to zero the $(t+1)$-th coordinate of the vector $x_j^t$ (and $\Delta_j^t$); and in the standard basis this coordinate can be calculated as $\langle x_j, o_i^t \rangle$. Thus, we want to maximize over $i$:

$$\sum_{j=1}^{M} \langle x_j, o_i^t \rangle^2 = \sum_{j=1}^{M} o_i^{tT} x_j x_j^T o_i^t = o_i^{tT} \left( \sum_{j=1}^{M} x_j x_j^T \right) o_i^t = o_i^{tT} X^T X o_i^t.$$

The procedure is summarized in Algorithm 1.

---

**Algorithm 1** $l_2$-greedy support items selection

compute $X^T X$
compute normalized vectors $o_i^0 = x_i / \|x_i\|_2$
**for** $t$ in $[1, \ldots, m]$ **do**
    choose $i_{t+1}$ that maximize $o_i^{tT} X^T X o_i^t$
    **for** j in $[1, \ldots, M]$ **do**
        $o_j^{t+1} \leftarrow o_j^t - o_{i_{t+1}}^t \langle o_{i_{t+1}}^t, o_j^t \rangle$
        $o_j^{t+1} \leftarrow o_j^{t+1} / \|o_j^{t+1}\|_2$
    **end for**
**end for**

---

The choice of the next support item may be trivially implemented with $O(n^2 M)$ complexity. But it can be optimized: together with $o_j^t$ we can keep the vectors $c_j^t = X^T X o_j^t$ that can be computed once initially in $O(n^2 M)$ and can be updated at each iteration synchronously with $o_j^t$. Updates of $o_j^t$ at each iteration have the form $o_j^{t+1} = \alpha_j o_j^t + \beta_j o_{i_{t+1}}^t$, so $c_j$ transforms analogously with the same coefficients: $c_j^{t+1} = \alpha_j c_j^t + \beta_j c_{i_{t+1}}^t$. So we can score all the items in $O(nM)$, calculating all the dot products $\langle o_j^t, c_j^t \rangle$, and update the vectors $o_j$ and $c_j$. The total complexity of the algorithm is $O(nM(n+m))$.

## C. Datasets

**ZESHEL**   The Zero-Shot Entity Linking (ZESHEL) dataset was constructed by Logeswaran et al. (2019) from Wikia. The task of zero-shot entity linking is to link mentions of objects in the text to an object from the list of entities with related descriptions. The dataset consists of 16 different domains. Each domain contains disjoint sets of entities, and during testing, mentions should be associated with entities solely based on entity descriptions. We run the experiments on five domains from ZESHEL selected by Yadav et al. (2022). As a heavy ranker $R$, we use the publicly available cross-encoder trained by Yadav et al. (2022).

*Table 6.* Datasets sizes.

|          | items | queries (used) |
|----------|-------|----------------|
| Yugioh   | 10031 | 3374 |
| P.Wrest. | 10133 | 1392 |
| StarTrek | 34430 | 4227 |
| Dr.Who   | 40281 | 4000 |
| Military | 105K  | 2400 |
| RecGames2| 16514 | 6958 |
| RecGames1| 16514 | 6958 |
| RecMusic | 8950  | 10179 |
| QA.Small | 82326 | 9650 |
| QA       | 0.8M  | 9650 |

**Question-Answering**   To additionally cover the question-answering domain, we conducted experiments on the MS MARCO (Bajaj et al., 2016) based dataset provided by Hugging Face.[8] As a heavy ranker, we use *all-mpnet-base-v2* from the SentenceTransformers library (Reimers & Gurevych, 2019), which is trained, among other datasets, on the MS MARCO data. We took ∼10K test queries and 0.8M passages corresponding to them (QA). For the experiments in Table 1, we used the smaller (QA.Small) version with 82K passages (only the test passages).

**RecGames**   We collected a dataset from a production service Yandex Games providing recommendations of games to users. Here, a heavy ranker $R$ is the CatBoost gradient boosting model (Prokhorenkova et al., 2018) trained on a wide range of features, including categories and other static attributes of items, social information (age, language, etc.) of users, item and user statistics, real-time statistics of user and item interactions, features derived from the matrix factorizations and multiple two-tower neural networks that use the features listed above as their features. There are two versions of this dataset. In RecGames1, CatBoost was trained to predict the time that a user spends on a given item after the click (in one session). In RecGames2, CatBoost was trained to predict the item with the longest time spent for some long period of time after the click (including new sessions).

**RecMusic**   To further increase diversity of the considered domains, we collected a dataset from another production recommendation service — Yandex Music. Here, CatBoost is used as a heavy ranker, and a dual encoder is a baseline. Both CE and DE are trained on a large set of external data and features. A smaller subsample of this data was used to train RBE, since we believe that RBE is able to show good quality even on a significantly smaller size of the training data.

Table 6 shows the statistics of all the datasets. Note that DEs were trained on a significantly larger number of queries, the table refers to the data used for AnnCUR/RBE training.

## D. Details on RBE Implementation

In this section, we discuss our implementation of the relevance-based embeddings. The code is available at https://github.com/shevkunov/Relevance-Based-Embeddings-Lightweight-Candidate-Retrieval.

As a trainable mapping $f_Q(R(S_I, q), \theta_Q)$, we use the following variant:

$$f_Q(R(S_I, q), \theta_Q) := R(S_I, q) \,||\, F_Q^{mlp}(R(S_I, q), \theta_Q) \,||\, (1),$$

where $F_Q^{mlp}$ is a 2-layer perceptron with the ELU activations, $||$ is the vector concatenation, and the last term is needed to represent the item-specific biases as a scalar product. The intuition here is that we split the representation into the prediction of AnnCUR and the trainable prediction of its error. In the experiments, such decomposition improves the convergence and training stability.

---

[8]v1.1 version from https://huggingface.co/datasets/microsoft/ms_marco.

*Table 7.* DE vs RBE on RecGames1 (extended).

| $X$ | Dual Encoder | AXN$_{\text{DE}}$ | RBE+KMeans | RBE+$l_2$-greedy |
|---|---|---|---|---|
| | HR$(X+100, X)$ | HR$(X+100, X)$ | HR$(X, X)$ | HR$(X, X)$ |
| 100 | 0.7048 | **0.7065** | 0.6300 | 0.6682 |
| 200 | 0.6803 | 0.6769 | 0.6611 | **0.6955** |
| 300 | 0.6739 | 0.6740 | 0.6912 | **0.7221** |
| 400 | 0.6739 | 0.6769 | 0.7109 | **0.7406** |
| 500 | 0.6760 | 0.6820 | 0.7253 | **0.7538** |
| 600 | 0.6792 | 0.6883 | 0.7357 | **0.7639** |
| 700 | 0.6827 | 0.6958 | 0.7448 | **0.7720** |
| 800 | 0.6868 | 0.7054 | 0.7527 | **0.7792** |
| 900 | 0.6904 | 0.7161 | 0.7589 | **0.7853** |
| | HR$(X+100, 100)$ | HR$(X+100, 100)$ | HR$(X, 100)$ | HR$(X, 100)$ |
| 100 | 0.7048 | **0.7065** | 0.6300 | 0.6682 |
| 200 | 0.7977 | 0.7970 | 0.8090 | **0.8359** |
| 300 | 0.8518 | 0.8538 | 0.8823 | **0.9026** |
| 400 | 0.8855 | 0.8902 | 0.9190 | **0.9342** |
| 500 | 0.9086 | 0.9153 | 0.9402 | **0.9522** |
| 600 | 0.9258 | 0.9331 | 0.9536 | **0.9632** |
| 700 | 0.9385 | 0.9465 | 0.9629 | **0.9704** |
| 800 | 0.9484 | 0.9572 | 0.9691 | **0.9760** |
| 900 | 0.9561 | 0.9660 | 0.9740 | **0.9799** |

For the item mapping $f_I(R(i, S_Q), \theta_I)$, we use the following function:

$$
\begin{aligned}
f_I(R(i, S_Q), \theta_I) &:= t_I(R(i, S_Q), \theta_I) \, || \, F_I^{mlp}(t_I(R(i, S_Q), \tilde{\theta}_I)) \, || \, (c_i), \\
t_I(R(i, S_Q), \theta_I) &:= R(i, S_Q) \times P, P = \text{pinv}(R(S_I, Q_{train})), \\
\theta_I &:= (P, c, \tilde{\theta}_I),
\end{aligned} \tag{9}
$$

where $c$ is a trainable bias vector, $\tilde{\theta}_I$ — perceptron trainable parameters. Although, as noted in Section 3.5, the transformation $f_I$ acts in practice on a finite set $I$ of elements and can be learned as an embedding matrix, the approach described above greatly accelerates the speed and stability of learning.

The mappings are trained using the Adam algorithm to optimize the following loss function inside the batches (both items and queries could be sampled):

$$
L := \frac{\sum\limits_{q \in Q_{train}} \text{softmax}(\tilde{R}(q, I)) \cdot (2 \cdot \mathbb{1}_{\text{binRelevance}}(q) - 1)}{|Q_{train}|},
$$

$$
\text{binRelevance}(q) := \tilde{R}(q, I) \geq q_{1 - \frac{K}{|I|}}(R(q, I)),
$$

where $K$ is the desired top size and $q_x(v)$ calculates the $x$-th quantile of the vector $v$. We have experimented with various loss functions, but the one described above leads to consistently good results.

Our implementation of RBE has about 50K trainable parameters.

## E. Additional Experiments

### E.1. Dual Encoder Embeddings vs Support Relevances (Extended)

Table 7 is the extended version of Table 4 from the main text.

### E.2. Additional Experiments on QA

*Table 8.* QA sample, different heavy ranker.

| $X$ | Dual Encoder | AnnCUR | AnnCUR+KMeans | AnnCUR+$l_2$-greedy | RBE+$l_2$-greedy |
|-----|--------------|--------|---------------|---------------------|------------------|
|     | HR($X+100, 100$) | HR($X, 100$) | HR($X, 100$) | HR($X, 100$) | HR($X, 100$) |
| 100 | **0.5444** | 0.4148 | 0.4511 | 0.4983 | 0.5417 |
| 200 | 0.6218 | 0.5571 | 0.5962 | 0.6491 | **0.6920** |
| 300 | 0.6721 | 0.6319 | 0.6627 | 0.7137 | **0.7556** |
| 400 | 0.7079 | 0.6778 | 0.7034 | 0.7517 | **0.7928** |
| 500 | 0.7350 | 0.7096 | 0.7318 | 0.7775 | **0.8186** |
| 600 | 0.7565 | 0.7339 | 0.7532 | 0.7966 | **0.8379** |
| 700 | 0.7741 | 0.7531 | 0.7704 | 0.8120 | **0.8528** |

We additionally conducted an experiment using the ms-marco-MiniLM-L-6-v2 model[9] as a heavy ranker. Since this model is heavier, we used only 10K passages from the QA dataset. We also compare HitRate($X+100, X$) for all-mpnet-base-v2 as DE with HitRate($X, X$) for RBE (DE has a larger re-ranking budget due to 100 support items relevance computations of RBE).

We note that this comparison may underestimate RBE, since the RBE embedding dimension is 100, whereas the DE embedding dimension is 768.

### E.3. Category Prediction from Relevance-Based Embeddings

Often in practice, embeddings trained to solve one problem are also applied to other ones. Following this, we conducted an additional experiment on predicting the categories of items in the RecGames1 dataset and obtained that the relevance vectors are informative vector representations for this task.

Namely, we trained a simple MLP to predict the category from either the CUR-transformed relevance vectors $R(I, S_Q) \times \text{pinv}(R(S_I, S_Q))$ obtained with different support items selection strategies or DE embeddings (sizes are the same). The categories of the elements are marked by the authors of the content among 30 available options, multiple categories are allowed. Intersection over Union (IoU) is used as a metric:

$$\text{IoU}(C_{pred}, C_{true}) = \frac{C_{pred} \cap C_{true}}{C_{pred} \cup C_{true}}.$$

The results are presented in Table 9. Two conclusions can be made: first, in this setup, categories are predicted better by vectors derived from relevance than by DE vectors; and second, selecting better support items for the relevance retrieval task also improves the quality of category prediction. This experiment demonstrates that the idea of representing elements using relevance vectors provided by a complex model has potential beyond relevance retrieval. Although these results are promising, we note that the possibility of such a transfer in other tasks and data requires a more detailed study.

*Table 9.* Category prediction on RecGames1.

| Support selection | IoU |
|-------------------|-----|
| (Dual Encoder) | 0.6796 |
| Random | 0.7331 |
| KMeans | 0.7419 |
| $l_2$-greedy | 0.7516 |

### E.4. Standard Deviation for AnnCUR (with Random Selection)

Since our baseline includes a random selection of support elements, the results can be noisy. In Table 10, we provide the average and standard deviation of HitRate(100) value (similar to Table 1), obtained by aggregating 15 runs of the algorithm with different initializations. As shown, even a random selection of support items gives fairly stable results. Moreover, it can be noted that the results of most non-random selection and RBE algorithms are several standard deviations higher.

### E.5. Selecting Support Elements by Popular Strategy

The results for AnnCUR + Popular are shown in the third row of Table 1, the results for RBE + Popular for the ZeSHEL datasets are shown below in Table 11.

---

[9] https://huggingface.co/cross-encoder/ms-marco-MiniLM-L6-v2

*Table 10.* Average and standard deviation for AnnCUR over 15 launches.

|     | Yugioh | P.Wrest. | StarTrek | Dr.Who | Military | RecGames2 | RecGames1 | RecMusic | QA.Small |
|-----|--------|----------|----------|--------|----------|-----------|-----------|----------|----------|
| avg | 0.4599 | 0.4232   | 0.2447   | 0.1875 | 0.2557   | 0.6747    | 0.5908    | 0.1492   | 0.5803   |
| std | 0.0081 | 0.0062   | 0.0118   | 0.0060 | 0.0037   | 0.0048    | 0.0087    | 0.0066   | 0.0030   |

*Table 11.* Popular as support items selection; HitRate(100) is reported (larger is better).

|                           | Yugioh | P.Wrest. | StarTrek | Dr.Who | Military |
|---------------------------|--------|----------|----------|--------|----------|
| AnnCUR + Popular (Table 1) | 0.2429 | 0.3001   | 0.1154   | 0.1197 | 0.1907   |
| RBE + Popular              | 0.2637 | 0.3161   | 0.1330   | 0.1207 | 0.2038   |

## F. Scalability Hints

Let us consider separately the selection of the support elements, training, and inference.

**Support items selection**    Since different clustering approaches have shown near-optimal quality, there are different options for scalable clustering:

- Clustering on downsampled datasets: for extremely large (and dense) datasets it is natural to expect that cluster structure could be inferred from a significantly smaller subsample (according to the authors' experience, on the data of ads recommender systems with billions of banners and downsampling to millions, this is so). Even for our research (small) datasets, downsampling 75% of the data does not lead to very significant performance drops (Table 12).

- Using data-driven clusterization: as shown in the third row of Table 1 (for the RecGames datasets), choosing popular items from different categories/genres (in our case, the global top of popular items is almost uniformly diversified) works extremely well.

- Using a distributed clustering algorithm.

*Table 12.* Support items selection on downsampled data; HitRate(100) is reported (larger is better).

|                        | Yugioh | P.Wrest. | StarTrek | Dr.Who | Military |
|------------------------|--------|----------|----------|--------|----------|
| AnnCUR+Random          | 0.4724 | 0.4280   | 0.2287   | 0.1919 | 0.2455   |
| AnnCUR+KMeans          | 0.5083 | 0.4850   | 0.3226   | 0.2517 | 0.3042   |
| AnnCUR+1/4KMeans       | 0.5112 | 0.4685   | 0.3101   | 0.2514 | 0.2854   |
| AnnCUR+1/8KMeans       | 0.5111 | 0.4694   | 0.3103   | 0.2450 | 0.2950   |
| AnnCUR+$l_2$-greedy    | 0.5618 | 0.5119   | 0.3677   | 0.2960 | 0.3357   |
| AnnCUR+1/4$l_2$-greedy | 0.5529 | 0.4865   | 0.3582   | 0.2862 | 0.3270   |
| AnnCUR+1/8$l_2$-greedy | 0.5492 | 0.4712   | 0.3533   | 0.2822 | 0.3231   |

**Training**    The training of RBE does not significantly differ from any dual-encoder-like models (which are commonly used in production recommendation services) with relevance vectors as inputs. The only major difference is that relevances to fixed support items should be provided. Let us also note that efficient sampling of negative samples for the loss function should be used in order to train on such large datasets.

**Inference**    Inference is also similar to dual-encoders: the item representations are precomputed and placed in the Approximate Nearest Neighbors index like HNSW, which accepts the embedding of the query as input.

