# OpenReview forum: "Relevance-Based Embeddings: Lightweight Candidate Retrieval via Heavy-Ranker Calls"
_ICML.cc/2026/Conference — ICML 2026 regular_

### Official Review · Reviewer_xQYA · 2026-02-26

**Soundness:** 3
**Presentation:** 3
**Significance:** 3
**Originality:** 4
**Overall Recommendation:** 5
**Confidence:** 3

**Summary:**

Overall, this paper introduces a novel Relevance-Based Embeddings approach that offers a balance between retrieval accuracy and efficiency. The authors theoretically prove that the method can consistently approximate arbitrarily complex continuous correlation functions, and propose a support term selection method based on a greedy strategy.

**Compliance With Llm Reviewing Policy:**

Affirmed.

**Key Questions For Authors:**

A major concern is that during real-time inference, in order to generate the embedding of a query, the system must first call CE to calculate the score of the query with m supporting items. Is this an acceptable latency cost in a real online service? Quantitative experiments and discussions would be helpful.

**Limitations:**

A detailed limitation part is lacking.

**Strengths And Weaknesses:**

Strengths:
1. The idea of ​​explicitly preserving the scoring information of key support terms of complex models through a more lightweight RBE is novel and reasonable.
2. Solid theoretical foundation. The authors proved that in a compact topological space, any continuous correlation function can be uniformly approximated by RBE with arbitrary precision.
3. Extensive experiment show that the proposed method outperforms existing baselines.

Weaknesses:
1. To generate the query's embedding, the system must first call a time-consuming reordering model to calculate its score against the supporting items, which is a potential weakness in systems with low latency requirements.
2. The upper limit of RBE's capabilities is highly tied to the reorderer model, which means that if the model itself has biases, embedding will directly inherit or even amplify these defects.

---

> ### Author Rebuttal · Authors · 2026-03-31
>
> Thank you for your review, insightful comments and positive feedback regarding our work! Below we reply to the raised concerns.
>
> **W1 & Q1**. We indeed need to use the heavy ranker calls to compute the query embedding. In our experiments, we take it into account, and all the compared methods are aligned and use the same number of heavy ranker calls. Indeed, at the final stage, all the algorithms use the scores of the heavy ranker to obtain the final list of items, and this step is crucial for obtaining an accurate list. Since the proposed algorithm uses several (100 in our experiments) heavy ranker calls to obtain query representations, we reduce the budget for the final re-ranking accordingly. Despite this reduction, we still obtain better accuracy of the final list. This demonstrates that using heavy ranker calls allows us to obtain significantly better query representations, which compensate for the re-ranking budget reduction. Thus, the obtained method is not more expensive at inference time.
>
> **W2.** Thank you for this comment, this is a very reasonable concern. If the final ranking model is not strong enough, better pre-selection approaches may lead to worse user experience, which applies not only to the proposed RBE but to pre-selection strategies in general. However, let us note that a strong and reliable final ranking model is a key ingredient for any recommender service, and without that, a reasonable final performance cannot be guaranteed. That is why such models are often computationally expensive, which served as the motivation for our study. Following your comment, we will add this discussion to the limitations section of the paper.
>
> If you have any additional questions or comments, we are happy to discuss them.

---

> > ### Author Rebuttal · Reviewer_xQYA · 2026-04-01
> >
> > Based on the author's feedback, my concerns have been resolved.

---

### Official Review · Reviewer_wino · 2026-03-13

**Soundness:** 2
**Presentation:** 2
**Significance:** 2
**Originality:** 3
**Overall Recommendation:** 4
**Confidence:** 5

**Summary:**

The paper introduces a method to train a retriever efficiently, when a stronger "heavy" reranking model is available. The reranker model can have arbitrary structure.
The main idea is borrowed from CUR where each document is represented as its relevance to a support set of queries, and vice versa. Paper shows the importance of selection of these support vectors. After evaluating relevance to corresponding support vectors for queries and docs, the paper learns a small network to map both to a shared space which is used for eventual retrieval. Paper shows results on moderately sized, non-standard retrieval benchmarks and shows modest gains against much bigger dual encoder baselines.

**Compliance With Llm Reviewing Policy:**

Affirmed.

**Final Justification:**

Authors addressed some of my concerns (comparison with modern baselines, cross encoder rankers on open datasets) through the rebuttal discussion, some concerns around the usefulness of the general direction which this paper works in and its comparison with dual encoders is not given full justice.

**Key Questions For Authors:**

Look at weaknesses. My main concerns are around
1. Did you evaluate distillation as a baseline?
2. Did you evaluate on bigger and more standard evaluation benchmarks? How do state of the art embedding models do on your datasets, where you can evaluate them?
3. Did you do cost/ latency comparison across baselines and your method? Calling heavy ranker model 100 times per query seems expensive before retrieval.

**Limitations:**

yes

**Strengths And Weaknesses:**

# Strengths
The idea of using existing reranker models and lightweight training to build a retrieval model is interesting, specially because the method needs less data. The paper beats DE baselines with many more parameters with just 50k trainable parameters. Also, the expirical reuslts around support selection is good. l2-greedy and clustering strategies beat random selection approaches.

# Weaknesses

- The formulation is expensive at inference time as compared to simple DE models because it invokes the strong reranker model ~100 times.
- The reranking models used in experiments are weak. Specially, for QA tasks all-mpnet-base-v2 is a very small dual encoder model, which beats the entire premise of the paper. The paper supports their method by pairwise features in rerankers, using dual encoders as heavy rerankers defeats that argument.
- State of the art DE models are much bigger than the ones evaluated in the paper which paper does not evaluate. These also generalize to corpuses they are not trained on. It is unclear whether the method in the paper would generalize beyond the corpus it is trained on, and whether one would need to train on all corpuses. Zero-shot evaluation on benchmarks like BEIR would be useful.
- Distillation like methods are a potential other baseline that solves the same problem. Rather than relying on support vectors at inference time, one can distil the query-document reranker scores into a smaller 50k parameter dual encoder model like TwinBERT (CIKM 2020) like approach initialized with a random backbone. This baseline is not evaluated, and the dismissal that "still requires two heavy models" is incorrect.
- The benchmarks that paper evaluates on a non-standard. Even for MS-MARCO why was evaluation not done on the full original dataset? BEIR/ MTEB are more popular and standard benchmarks for retrieval. ZESHEL datasets are also small. Evaluations on larger datasets would have given more confidence in scalability of proposed approach.

---

> ### Author Rebuttal · Authors · 2026-03-31
>
> Thank you for your detailed and thoughtful review. Below we reply to the raised concerns.
>
> **W1 & Q3.** Please note that in the experiments, we take the additional heavy ranker calls into account, and **all the compared methods use the same number of heavy ranker calls**. At the final stage, all the algorithms use the heavy ranker scores to obtain the final list of items, and this step is crucial for obtaining an accurate list. Since the proposed algorithm uses several (100 in our experiments) heavy ranker calls to obtain query representations, we reduce the final re-ranking budget accordingly. Despite this reduction, we still obtain better accuracy of the final list. This demonstrates that using heavy ranker calls allows us to obtain **significantly better query representations**, which compensate for the re-ranking budget reduction. Thus, the obtained method is not more expensive at inference time.
>
> To better address the remaining concerns, let us describe our motivation behind the chosen datasets and relevance models. When we planned the experiments, we wanted to find out whether relevance vectors are powerful query and item representations that can be used in a variety of tasks when an arbitrary relevance function for query-item pairs is available. Thus, we aimed to cover scenarios that are diverse in several aspects:
>
> - **Tasks:** entity linking, question answering, recommendations;
> - **Relevance models:** different heavy rankers, including neural networks and gradient boosting;
> - **Setups:** both reproducible academic benchmarks and in-the-wild production systems with strongly tuned dual encoders.
>
> **W2.** As described above, we’ve chosen various types of heavy rankers to check the generalizability of the proposed approach. Please note that for the three RecSys datasets, the heavy rankers are strong models that are used in production and rely on pairwise features. For ZESHEL, we compare our results with the approach by Yadav et al. (2022), which outperforms CE models.
>
> **W4 & Q1.** In the experiments on the three RecSys datasets, we compared our approach with powerful dual encoders: these are the best dual encoders available in the corresponding production services, and the development of such methods included experiments with distillation approaches. We conducted these experiments to make sure that the proposed approach can successfully compete with the strongest available DEs trained on all the available data.
>
> Also, like all dual encoders, distilled models have a fundamental theoretical limitation: they rely significantly on the informativeness of individual features. RBE surpasses this limitation via several computations of the heavy ranker. For instance, in a hypothetical scenario when only the pairwise features are available, distillation-based models would not work, while the proposed approach works.
>
> Regarding the particular example with TwinBERT, if we are not mistaken, 50K is the vocabulary size, while the number of parameters is 35 million (according to Section 5.2 of the TwinBERT paper).
>
> **W5 & Q2.** Conducting comprehensive experiments requires significant computational resources: e.g., to evaluate the proposed approaches, we need to compute the exact list of the most relevant items, i.e., for each test query, we need to compute the heavy ranker scores for all the items. Also, in our experiments, we opted to evaluate the universality of the proposed approach considering diverse tasks, relevance models, and setups. Let us also note that in the RecSys datasets, the number of elements corresponds to a real production system. Despite its medium size, the complexity of the heavy ranker and the latency requirements of the service require efficient approximate retrieval of the relevant items.
>
> Regarding our choice of datasets, we started with ZESHEL since these datasets have been used by Yadav et al. (2022), which is the work most relevant to our study. Then we added recommendation datasets from real production services to check how the method works “in the wild” and against strong dual encoders. The MS-MARCO dataset was added to test how our approach scales to larger datasets; we’ve chosen this version since it’s already noticeably larger than our other datasets.
>
> We strongly believe that our experimental setup maintains a good balance between the diversity of the covered setups and reasonable computational resources for the research project. The experiments we conducted convinced us of the usefulness of relevance-based embeddings, and we sincerely believe that the idea is useful for the ML community. That said, we are open to suggestions. If there is a particular experiment that would convince you of the usefulness of the proposed approach, please let us know, and we will do our best to provide the results during the discussion period. If this is the case, please let us know which dataset, heavy ranker, and, if available, dual encoder baselines to consider.

---

> > ### Author Rebuttal · Reviewer_wino · 2026-04-01
> >
> > Thank you authors for your clarifications.
> >
> > Thank you for clarifying your benchmark selection criteria. I would have liked more coverage on more standard benchmarks, but the empirical treatment given to the existing benchmarks is also subpar.
> >
> > The paper provides essentially NO verifiable empirical evidence that RBE is better than DE (dual encoder) approaches. Public datasets are only used in Tables 1 and 2, where comparison against DE is lacking.
> >
> > Why did you not:
> > - Use an actual cross-encoder as the heavy ranker on MSMARCO? The paper uses all-mpnet-base-v2, which is a bi-encoder dual encoder, not a cross-encoder. Publicly available cross-encoders trained on MSMARCO exist (like ms-marco-MiniLM-L-6-v2 or BAAI/bge-reranker-base). Using a bi-encoder as the "heavy ranker" undermines the paper's core narrative.
> > - On ZESHEL, compare against more recent methods? AnnCUR (Yadav et al., 2022) is no longer the state of the art. ADACUR (Yadav et al., 2023) and AXN (Yadav et al., 2024, ICLR) both substantially improve upon AnnCUR on ZESHEL with the same CE. CMC (EMNLP 2024) also reports strong results. Moreover, the standard metric on ZESHEL is Recall@k which is used by all of the above papers, not HitRate, which prevents any comparison with broader literature.
> >
> > For TwinBERT comparison, I meant designing a simple architecture and training it with distillation apples-to-apples. It's unclear whether the DE baselines evaluated on RecSys datasets were trained under a controlled setup. One could design a simple architecture with ~50K parameters. For example, projected BERT features with a few-layer MLP on top, to give a fair parameter-matched comparison. The core point I am making is that with no controlled and trained DE baselines on private RecSys datasets, accuracy claims against DE methods don't have weight. The original AnnCUR paper had controlled DE ablations on their setups and deeper evaluation (e.g., comprehensive Recall@k curves at multiple operating points across ZESHEL domains).
> >
> > The paper's foundation is on better support selection and neural embeddings for the AnnCUR approach from Yadav et al. (2022), but the empirical treatment (and shaky at best claims against DE methods) does not meet the bar for an ICML paper for me.

---

> > > ### Author Response · Authors · 2026-04-08
> > >
> > > Thank you for your detailed reply! In this final response we would like to clarify some points.
> > >
> > > > ​​Public datasets are only used in Tables 1 and 2, where comparison against DE is lacking.
> > >
> > > The experiments on the ZESHEL datasets are conducted following the setup of Yadav et al. (2022). In particular, our HitRate$(k, k_r)$ is equivalent to Top-$k$-Recall@$k_r$ from Yadav et al. (2022). When testing our AnnCUR implementation, we verified that the results agree with Figures 10-24 in Yadav et al. (2022). As Yadav et al. (2022) conducted a broad comparison of AnnCUR with various dual encoders, we refer to their comparison and focus on the detailed analysis of the proposed approach and AnnCUR. If required, we can re-run the dual encoders from Yadav et al. (2022) and add the results to our tables for better readability. Since we follow their setup, we expect similar results.
> > >
> > > >  Use an actual cross-encoder as the heavy ranker on MSMARCO?
> > >
> > > We conducted preliminary experiments on MSMARCO sample with the suggested CE. In our setup (but with this new relevance function) the results are:
> > >
> > > | | |
> > > | --- | --- |
> > > | AnnCUR |  0.3998 |
> > > | AnnCUR + KMeans | 0.4399 |
> > > | AnnCUR + l2-greedy |  0.4662 |
> > > | RBE + l2-greedy | 0.4750 |
> > >
> > > We also compare HitRate(300, 200) for DE with HitRate(200, 200) for RBE (since DE has a larger re-ranking budget):
> > >
> > > | | |
> > > | --- | --- |
> > > | RBE + l2-greedy | 0.5662 |
> > > | DE | 0.4884 |
> > >
> > > The results for RBE are currently a weaker estimation due to smaller embedding size (100 for RBE vs 768 for DE) and shorter training time (response period limitations). We will add complete and extended results with detailed descriptions to the updated paper.
> > >
> > > > On ZESHEL, compare against more recent methods? AnnCUR (Yadav et al., 2022) is no longer the state of the art. ADACUR (Yadav et al., 2023) and AXN (Yadav et al., 2024, ICLR) both substantially improve upon AnnCUR on ZESHEL with the same CE.
> > >
> > > ADACUR selects support items individually for each query, which leads to significantly increased query time (it becomes linear in the number of elements), which makes the approach infeasible in most practical applications (see line 122). Hence, we only compared with AXN that is more recent and was shown to outperform ADACUR.
> > >
> > > AXN is an interesting approach that predicts the difference between DE and CE predictions at the query time by computing some CE scores. There are two limitations: the performance of AXN strongly depends on a reliable DE and it has increased query time due to the need for iterative refining all item relevances for each query.
> > >
> > > We use AXN as one of our baselines on three RecSys datasets (Tables 4, 5, 7), where we have strong dual encoders to build AXN upon.
> > >
> > > > the standard metric on ZESHEL is Recall@k which is used by all of the above papers, not HitRate, which prevents any comparison with broader literature.
> > >
> > > As we mentioned above, our HitRate$(k, k_r)$ is equivalent to Top-k-Recall@$k_r$ from Yadav et al. We will add a comment to the paper to clarify that.
> > >
> > > > with no controlled and trained DE baselines on private RecSys datasets, accuracy claims against DE methods don't have weight.
> > >
> > > We do not agree that our experiments on real production datasets do not have weight. We use both academic and private experiments since not all realistic scenarios can be reproduced with open benchmarks (e.g., due to privacy reasons or due to complexity of experimental pipelines of real production systems). In our case, the amount of data on which both the CE and DEs are trained go way beyond the basic dataset that we use for training our model. This provides us with the strongest possible models. While we plan to share the dataset used for training RBE and AnnCUR (and also share the DE embeddings), we understand that it will not allow for reproducing the training of CEs and DEs since they have been trained on various external data. That is why we use this realistic setup together with open datasets.
> > >
> > > > The original AnnCUR paper had controlled DE ablations on their setups and deeper evaluation
> > >
> > > We agree that the AnnCUR paper provided deep evaluation of DE approaches. That is exactly why we rely on their results and focus on their best approach (AnnCUR) and our modifications (on the same datasets with the same metric). As written above, if needed, we can add the corresponding results for the DE approaches to our tables and we expect the results to agree with Yadav et al. (2022).
> > >
> > > Finally, we would like to empathise that an important part of our paper is the theoretical analysis that provides intuition about why relevance vectors are expressive and also shows that in some cases they are provably better than dual encoders. We analyze both the original AnnCUR and the proposed RBE. More generally, our analysis provides theoretical justification for various methods based on CE computations for a query and some support items since it shows the expressive power of such relevance vectors.

---

### Official Review · Reviewer_7mxL · 2026-03-13

**Soundness:** 3
**Presentation:** 4
**Significance:** 2
**Originality:** 3
**Overall Recommendation:** 5
**Confidence:** 3

**Summary:**

This paper studies a concept the authors call relevance-based embeddings. The core idea behind the paper is to model a retrieval setting by pre-selecting certain database items and measuring their relevance to a set of queries. This idea was previously studied in a series of papers, but these works all assumed that the pre-selected subset was chosen uniformly at random from the database collection. In this paper, the authors prove formally that one can improve relevance significantly by more carefully curating this subset, showing that simple but principled approaches can outperform a random strategy. The authors also empirically validate their claims on relevance and recommender systems benchmarks.

**Compliance With Llm Reviewing Policy:**

Affirmed.

**Final Justification:**

During the rebuttal process, the authors addressed my main questions regarding how the evaluation datasets were constructed and provided insights into which methods perform the best. I would encourage the authors to incorporate both of these additional points into the manuscript. Overall, the rebuttal process reinforced my prior assessment of the paper and I recommend acceptance.

**Key Questions For Authors:**

1. Do you have any insights into whether any particular support strategy might be the best? Do your results shed any light on this question. An insightful answer to this question would positively change my evaluation of the paper.

2. Can you provide more details on the datasets you evaluated on (i.e. how large are the datasets and why they were chosen)? I would positively change my evaluation of the paper with a detailed answer to this question.

**Limitations:**

Yes

**Strengths And Weaknesses:**

Soundness: The claims of the paper are well-supported by theoretical proofs and experimental results. The authors appropriately introduce the relevant prior work which they build upon and the proofs and assumption seem reasonable to me. The empirical experiments are also designed appropriately (though I would like the authors to describe the details of the benchmark datasets they used in more detail). The authors also provide a balanced assessment of their results, particularly with regards to the scalability of their method.

Presentation: The paper is very well written and structured where the ideas introduced in this work feels like a very natural extension of the previous work. I commend the authors on putting in the effort to communicate their findings clearly and succinctly.

Significance: This is the weakest aspect of the submission in my opinion. Although the authors' findings that more sophisticated sampling improves the quality of relevance-based embeddings, this insights, on its own, is not extremely surprising and I think a more impactful result might be to provide insight into what might be the *best* sampling strategy. In addition, the empirical lift over the previous best method is relatively modest in absolute terms. Nevertheless, I think the theoretical insights are the strongest aspect of the paper.

Originality: Yes, the work provides new insights and deepens our understanding of existing methods by analyzing the notion of relevance from a theoretical lens with new techniques.

---

> ### Author Rebuttal · Authors · 2026-03-31
>
> Thank you for your review, insightful comments and positive feedback regarding our work! We address your questions below.
>
> > 1. Do you have any insights into whether any particular support strategy might be the best? Do your results shed any light on this question.
>
> The support element selection strategies we’ve considered range from simple heuristics to the more advanced and theoretically justified approach that we call $l_2$-greedy. While even simple heuristics lead to noticeable improvements over the random selection, we see that $l_2$-greedy consistently outperforms all other approaches on most of the datasets (Table 1). From the theoretical perspective, this method is well justified as it constructs the set of support elements by directly optimizing the approximation errors (MSE). There are two aspects in which $l_2$-greedy can potentially be further strengthened: 1) greedy optimization can be suboptimal, 2) a particular application may require a specific quality measure for which the support element selection can be further adapted. We believe that in its current form, $l_2$-greedy provides a good balance of simplicity, universality, and good performance.
>
> > 2. Can you provide more details on the datasets you evaluated on (i.e. how large are the datasets and why they were chosen)?
>
> In our experiments, we wanted to show that relevance vectors are powerful query and item representations and that they can be used in a variety of tasks where an arbitrary relevance function for query-item pairs is available. Thus, we aimed to cover scenarios that are diverse in several aspects:
>
> - **Tasks:** entity linking, question answering, recommendations;
> - **Relevance models:** different heavy rankers, including neural networks and gradient boosting;
> - **Setups:** both reproducible academic benchmarks and in-the-wild production systems with strongly tuned dual encoders.
>
> We believe that this set of experiments convincingly demonstrates that the method is useful in diverse scenarios.
>
> Let us now discuss specific datasets that we use in the paper. The first set of datasets is ZESHEL, which was constructed by Logeswaran et al. (2019). We’ve chosen these datasets since they have been used by Yadav et al. (2022), which is the work most relevant to our study. The sizes of these datasets range from 10K to 105K items (see Table 6). Then, we extended this set of datasets by adding the QA dataset (0.8M items) and its smaller version. This allows us to increase the size of the considered datasets and also evaluate the approach in another domain. Finally, we added three recommendation datasets from real production services. With these datasets, we can check how the method works “in the wild” and, importantly, demonstrate that it can outperform even strong dual encoders trained on significantly larger amounts of data than those available for the proposed method. The statistics for all the datasets are listed in Table 6.
>
> We hope that we addressed the raised questions. If you have any additional questions and comments, we are happy to continue the discussion.

---

> > ### Author Rebuttal · Reviewer_7mxL · 2026-04-01
> >
> > The authors have addressed my two major questions. Given my initial support of the paper, I will keep my rating the same. My only additional suggestion to the authors is to call out that $l_2$-greedy consistently outperforms all other approaches more clearly in the manuscript.

---

### Official Review · Reviewer_ZZR6 · 2026-03-13

**Soundness:** 2
**Presentation:** 3
**Significance:** 3
**Originality:** 2
**Overall Recommendation:** 4
**Confidence:** 5

**Summary:**

Heavy ranker breaks the strict latency requirement in retrieval stage of cascade online systems (e.g. Recommender System, Information Retrieval). The paper tries to distill the knowledge of heavy ranker by Relevance-based embeddings. Concretely, queries (items) are represented by including the relation, rated by the heavy ranker, between queries(items) and the corresponding support item (query) set. The authors theoretically prove that the heavy ranker can be approximated by the inner product of the reformulated embeddings within arbitrary  error under some mild assumption.  The empirical studies verify the superiority of the proposed method.

**Compliance With Llm Reviewing Policy:**

Affirmed.

**Final Justification:**

I keep weak acceptance decision. Referring to the final discussion, the authors claim that the query can be computed immediately by calling the heavy relevance function. However, calling the heavy relevance function itself is time-consuming, which does not address my concern.

**Key Questions For Authors:**

None

**Limitations:**

refer to the weakness.

**Strengths And Weaknesses:**

Strengths:
1. Well organized and clear presentation.
2. The work inspires the approximate heavy ranker retrieval in cascade web-scale online application (e.g. Recommender system, Information Retrieval).
3. Provide the theoretical guarantee and comprehensive empirical studies to verity the effectiveness.

Weaknesses:
1. The theoretical result (e.g. Theorem3.2 ) lacks the trade-off analysis between the approximation error and the size of support set. If the the high precision requires the large support set, the approximation methods demonstrate less actual value.
2. In the experiment settings, the ANN search requires the pre-computation for the representation of queries and items. In real-world online system, the  item representation can be update asynchronously but the representation of queries need the real-time computation to extract the real-time intent of the query (e.g. the real-time interest of user in Recommender system).  This restriction hinders the practical deployment in real-world online system.

Overall, although some limitation, this paper can inspire more research about heavy ranker retrieval.

---

> ### Author Rebuttal · Authors · 2026-03-31
>
> Thank you for your review, insightful comments and positive feedback regarding our work! Below, we reply to the raised concerns.
>
> **W1.** Thank you for this comment, we agree that the analysis of the convergence rate is an important research topic that may bring further insights. Providing theoretical guarantees for the convergence rate is challenging given that the support elements can be chosen arbitrarily. Our experiments show that more advanced strategies for selecting support items significantly benefit the proposed approach. Following your comment, we will mention the analysis of the convergence rate as a promising research direction for future studies. In our work, we address this question empirically by showing that 1) the choice of support elements is important for better convergence, and 2) on various datasets considered in this paper, the proposed approach shows good performance compared to the baselines.
>
> **W2.** Regarding the real-time computation of query representations, we indeed need to use the heavy ranker calls for that. In our experiments, we take it into account, and all the compared methods are aligned and use the same number of heavy ranker calls in total. Indeed, at the final stage, all the algorithms use the scores of the heavy ranker to obtain the final list of items, and this step is crucial for obtaining an accurate list. Since the proposed algorithm uses several heavy ranker calls to obtain the query representation, we reduce the budget for the final re-ranking accordingly. Despite this reduction, we still obtain better accuracy of the final list. This demonstrates that using heavy ranker calls allows us to obtain significantly better query representations, which compensate for the re-ranking budget reduction.
>
> If you have any additional questions or comments, we are happy to discuss them.

---

> > ### Author Rebuttal · Reviewer_ZZR6 · 2026-04-01
> >
> > Thank you for your explanations regarding my concerns; while Weakness 1 (W1) is partially addressed through the empirical studies, Weakness 2 (W2) remains unaddressed. Especially regarding the infeasibility of query pre-computation in real-world online recommendation or information retrieval systems.  In such scenarios, real-time updates are essential to capture users' dynamically evolving intent at the moment a request is issued. For instance, in web search, the system cannot anticipate a user's query beforehand and thus cannot pre-compute query representations before the query is submitted, and similarly, in recommendation systems, user interests must be modeled on-the-fly as interactions occur rather than relying on offline pre-computation, a limitation that significantly hinders the practical deployability of the proposed method in industrial-scale, latency-sensitive applications. Typically, cached pre-computation (e.g., long-term user interests extracted from interaction history and updated daily) serves as the one of the inputs for real-time models in industrial scenarios.

---

> > > ### Author Response · Authors · 2026-04-05
> > >
> > > Thank you for your detailed response to our rebuttal! It appears that we initially misunderstood your concern regarding W2, and we would like to use this final response to clarify this point.
> > >
> > > Please note that in our experiments, query embeddings are computed in real-time; that is, each time a new query arrives, we compute its relevance to several items to obtain its embedding. These relevance scores take into account dynamically evolved query intent at the moment the query is issued. Moreover, they can be computed for previously unseen queries. The computed relevance is based on attributes that describe the individual query, the individual item, and the query-item pair, and these attributes can be both static or updated in real-time. Thus, in our scenario, both the candidate selection (based on the RBE embeddings) and the final re-ranking take into account the up-to-date information about the query.
> > >
> > > We hope that our response addresses your concern, and we are sorry for the initial misunderstanding.
> > >
> > > **Update after the final justification:**
> > >
> > > Thank you for your support! We would like to clarify the following points:
> > > - The proposed approach takes into account the up-to-date information, as written above.
> > > - We take into account the additional costs in our experiments, i.e., the number of heavy ranker computations is the same for all the models.
> > > - Cached pre-computation can be incorporated to the proposed approach, similarly to other methods.

---

### Decision · Program_Chairs · 2026-04-30

**Decision:**

Accept (regular)

**Comment:**

The paper studies the problem of computing lightweight representations of the data elements and queries in the first phase of  the “retrieve-and-rerank” approach to information retrieval. In this approach, a superset of relevant data items is first retrieved using a lightweight approach, e.g., by representing queries and items using embeddings, and then using fast approximate nearest neighbor data structures to retrieve the items. The superset is then re-ranked using an expensive re-ranker.  The paper proposes instead to represent queries and items by their relevance to a pre-selected set of queries or items (“support elements”). Although such methods have been investigated before by Morozov & Babenko, 2019; Yadav et al., 2022, the authors perform a systematic study of various methods for selecting support elements, showing that a significant improvement over SOTA is possible.

All reviewers ultimately recommended acceptance. During the rebuttal discussions, several concerns were raised, notably about experimental performance on other (more standard) benchmarks like MSMARCO, the feasibility of embedding queries in real time, lack of convergence analysis (e.g., in Theorem 3.2). The authors provided answers and additional experiments addressing those concerns. It is expected that the final version of the paper will incorporate those answers.

Altogether, a valuable addition to the vast literature on retrieve-and-rerank methods